# Integrative analysis of metabolite GWAS illuminates the molecular basis of pleiotropy and genetic correlation

Courtney J Smith[1]*[†], Nasa Sinnott-Armstrong[1,2]*[†], Anna Cichońska[3], Heli Julkunen[3], Eric B Fauman[4], Peter Würtz[3], Jonathan K Pritchard[1,5]*

[1]Department of Genetics, Stanford University School of Medicine, Stanford, United States; [2]Herbold Computational Biology Program, Fred Hutchinson Cancer Research Center, Seattle, United States; [3]Nightingale Health Plc, Helsinki, Finland; [4]Internal Medicine Research Unit, Pfizer Worldwide Research, Development and Medical, Cambridge, United States; [5]Department of Biology, Stanford University, Stanford, United States

**Abstract** Pleiotropy and genetic correlation are widespread features in genome-wide association studies (GWAS), but they are often difficult to interpret at the molecular level. Here, we perform GWAS of 16 metabolites clustered at the intersection of amino acid catabolism, glycolysis, and ketone body metabolism in a subset of UK Biobank. We utilize the well-documented biochemistry jointly impacting these metabolites to analyze pleiotropic effects in the context of their pathways. Among the 213 lead GWAS hits, we find a strong enrichment for genes encoding pathway-relevant enzymes and transporters. We demonstrate that the effect directions of variants acting on biology between metabolite pairs often contrast with those of upstream or downstream variants as well as the polygenic background. Thus, we find that these outlier variants often reflect biology local to the traits. Finally, we explore the implications for interpreting disease GWAS, underscoring the potential of unifying biochemistry with dense metabolomics data to understand the molecular basis of pleiotropy in complex traits and diseases.

**\*For correspondence:**
courtrun@stanford.edu (CJS);
nasa@fredhutch.org (NS-A);
pritch@stanford.edu (JKP)

[†]These authors contributed equally to this work

## Editor's evaluation

Smith and colleagues provide a framework for understanding a seemingly paradoxical observation in human genetics: two phenotypes may be closely correlated to each other, and the patterns of genetic variation that influence both phenotypes may be widely shared at the genome-wide level, but there are often specific genetic variants that show discordant patterns. Though the observations in this article are derived from analysis of metabolic phenotypes, this may have broader relevance to interpreting the results from disease-related genetic association studies, and shed light on the processes that connect different disease phenotypes.

## Introduction

A central challenge in the field of human genetics is understanding the mechanism of how genetic variants influence complex traits and diseases. Genome-wide association studies (GWAS) have begun characterizing the genetic architecture of complex traits, but the molecular mechanisms connecting genetic variants to these traits are rarely understood. This is particularly true for understanding pleiotropy, when a variant affects multiple traits (*Solovieff et al., 2013*). It is possible to estimate the genetic correlation between traits (*Bulik-Sullivan et al., 2015b*; *Shi et al., 2017*), but it is often unclear what contributes to this at a molecular

or physiological level. A handful of in vitro disease-focused 'post-GWAS' studies have convincingly shown the mechanisms driving pleiotropy of individual key associations (*Warren et al., 2017*; *Sinnott-Armstrong et al., 2021b*); however, these studies are highly specific and time-consuming. Developing statistical and computational approaches to identify putative molecular mechanisms is invaluable to advancing our understanding of where and how pleiotropic GWAS variants act.

In this study, we use metabolites as model traits to understand pleiotropic features of genetic architecture. Metabolites are small molecules interconverted by a series of biochemical pathways and are an appealing model system for studying pleiotropy because their pathways are typically well-documented and biologically simpler than those underlying other complex traits (*Gieger et al., 2008*; *Sinnott-Armstrong et al., 2021a*). Previous work in Mendelian genetics has identified inborn errors of metabolism (IEM) in many enzymes (*Woidy et al., 2018*). Metabolite GWAS, which have long observed pervasive pleiotropy at these IEM genes and other loci (*Shin et al., 2014*; *Yeung, 2021*), offer a potential opportunity to further explore the relationships between intermediate molecules and disease outcomes at scale. Here, we jointly analyzed GWAS results of 16 plasma metabolites from the Nightingale Health Nuclear Magnetic Resonance (NMR) Spectroscopy platform in nearly 100,000 individuals in the UK Biobank (*Julkunen et al., 2021*; *Figure 1*; see 'Methods'). These 16 metabolites included glucose, pyruvate, lactate, citrate, isoleucine, leucine, valine, alanine, phenylalanine, tyrosine, glutamine, histidine, glycine, acetoacetate, acetone, and 3-hydroxybutyrate. They were chosen based on their biochemical proximity to each other, their relevance to health and disease, and because the genes and enzymes involved in their metabolism are well-characterized. They play especially important roles in energy generation and energy storage pathways such as glycolysis, the citric acid cycle, amino acid metabolism, and ketone body formation. They are relevant to many metabolic diseases including type 2 diabetes (*Laffel, 1999*; *Guasch-Ferré et al., 2020*; *Newsholme et al., 2007*), cardiovascular disease (*Lusis and Weiss, 2010*), and non-alcoholic fatty liver disease (*Watt et al., 2019*).

Numerous GWAS have begun characterizing the genetic architecture of metabolites and found them to be heritable and polygenic (*Lemaitre et al., 2011*; *Suhre et al., 2011*). Recent metabolite studies have shown that leveraging information about the biochemical pathways relevant to a given metabolite (*Teslovich et al., 2018*; *Graham et al., 2021*; *Rueedi et al., 2017*; *Sinnott-Armstrong et al., 2021a*) can allow for more interpretable gene annotation of GWAS hits. This has led to the dissection of individual associations of biomarkers, such as lipids (*Kettunen et al., 2016*), glycine (*Wittemans et al., 2019*), and intermediate clinical measures (*Lotta et al., 2021*), with cardiometabolic and other diseases. The pervasive pleiotropy at these GWAS loci with other metabolites as well as disease (*Lotta et al., 2021*; *Pott et al., 2019*) suggests the potential of utilizing these data for investigating the mechanism of pleiotropic effects as a core component of genetic architecture. While recent GWAS have begun jointly investigating multiple metabolites (*Cichonska et al., 2016*; *Ruotsalainen et al., 2021*; *Qi and Chatterjee, 2018*), they have yet to do so in the context of their biochemical pathways.

In this article, we demonstrate that investigating the effects of pleiotropic variants on biologically related metabolites allows for a better understanding of why these variants have their observed joint effects. Our results reveal striking heterogeneity in genetic correlation across the genome and provide a biologically intuitive basis for understanding this heterogeneity. Together, this allows us to dissect the molecular basis of metabolic disease GWAS variants and enables us to directly define the mechanism relating an example variant to its associated disease.

## Results

### Insights into the shared genetic architecture of biologically related metabolites

We chose 16 metabolites from the 249 available through the Nightingale NMR platform in a subset of the UK Biobank (*Figure 1*; see 'Methods'). These 16 metabolites were selected based on their biochemical proximity, relevance to health and disease, and because the genes and enzymes involved in their metabolism are well-characterized. We classified the 16 metabolites into four groups based on shared biochemistry: Glycolysis (glucose, pyruvate, lactate, citrate), Branched Chain Amino Acid (BCAA; isoleucine, leucine, valine), Other Amino Acid (alanine, phenylalanine, tyrosine, glutamine, histidine, glycine), and Ketone Body (acetoacetate, acetone, 3-hydroxybutyrate). Trait measurements were log-transformed and adjusted for relevant technical covariates. After outlier removal, we obtained a primary dataset of 94,464 genotyped European-ancestry individuals with data for all 16 metabolites.

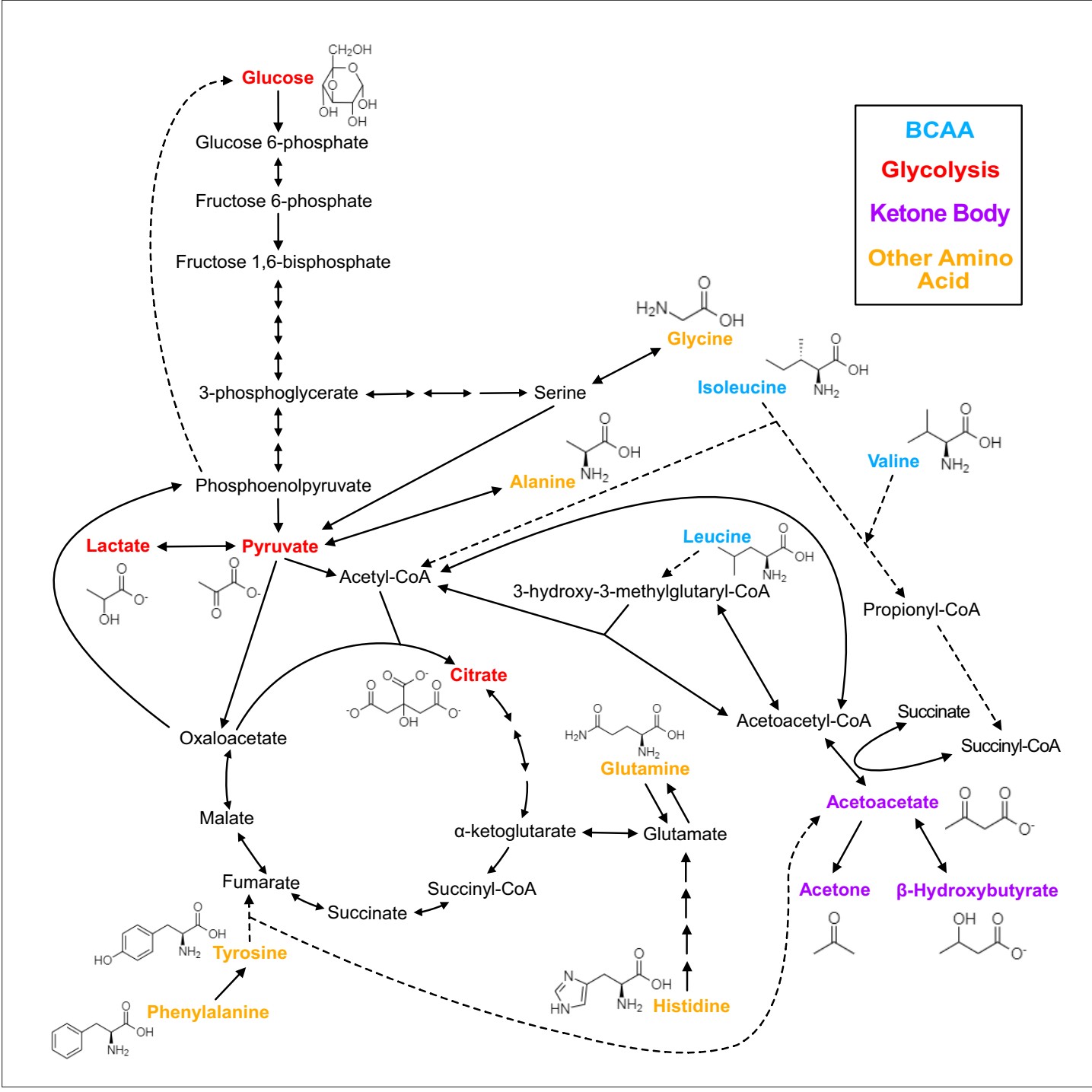

**Figure 1.** Biochemistry of relevant metabolites. Pathway diagram and molecular structure of relevant metabolites, colored by their biochemical groups. The pathway diagram was curated from multiple resources (see 'Methods'). All solid lines represent a single chemical reaction step. Dotted lines represent a simplification of multiple steps. For simplicity, only a subset of all the reactions each metabolite participates in is shown. Genes encoding the enzymes that catalyze the above chemical reactions are known and presented in *Figure 3—figure supplement 1*.

We first sought to characterize the genetic architecture underlying these metabolites by performing GWAS for each (*Figure 2—figure supplement 1*). Hits from individual GWAS were clumped with an $r^2$ of 0.01 per megabase, combined across metabolites, then pruned to the single nucleotide polymorphism (SNP) with the most significant p-value within 0.1 cM. This resulted in 213 lead variants with a genome-wide significant association in at least one metabolite, referred to as the metabolite GWAS

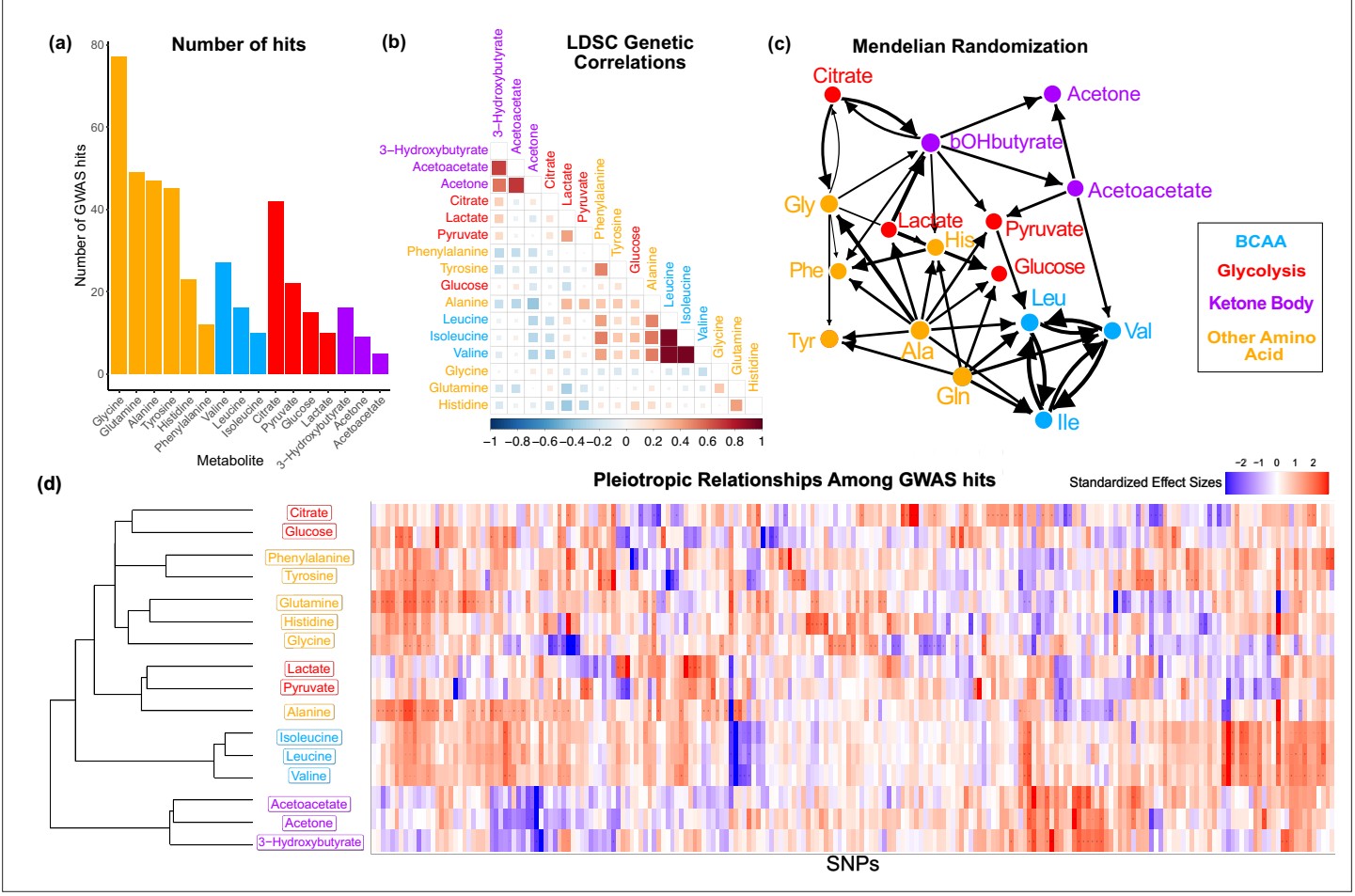

**Figure 2.** Overview genetic architecture of metabolites. (**a**) Number of genome-wide association studies (GWAS) hits per metabolite. (**b**) Pairwise LDSC genetic correlations between the metabolites, clustered by genetic correlation. (**c**) Mendelian randomization weighted results between the metabolites. (**d**) Biclustered standardized effect size in each metabolite for the 213 metabolite GWAS hits. For visualization, effect sizes were divided by the standard error then inverse normal transformed and standardized. Each variant was aligned to have a positive median score across metabolites.

The online version of this article includes the following figure supplement(s) for figure 2:

**Figure supplement 1.** Manhattan plots.

**Figure supplement 2.** HESS heritability plots.

**Figure supplement 3.** Phenotypic correlation.

**Figure supplement 4.** Quantile–quantile plots.

hits. Glycine had the largest number of significant associations with 77 hits (*Figure 2a*). There were 47 variants with significant associations in more than one metabolite, including rs2939302 (near the gene *GLS2*), which was significant in 9 of the 16 metabolites, and rs1260326 (*GCKR*), which was significant in 8. Glycine also had the highest Heritability Estimate from Summary Statistics (HESS) total SNP heritability of 0.284 (*Supplementary file 1* and *Figure 2—figure supplement 2*).

To understand the shared genetics of these metabolites, we then investigated the extent of pleiotropy between and within biochemical groups. In order to examine this, we first calculated pairwise LD Score regression (LDSC) genetic correlation across the 16 metabolites. We found substantial genome-wide sharing for many pairs of metabolites, especially for metabolites within the same biochemical group (*Figure 2b*; phenotypic correlation in *Figure 2—figure supplement 3*). We then explored pleiotropic effects beyond the polygenic background by examining the structure within the metabolite GWAS hits. Pairwise Mendelian randomization (MR) between the metabolites emphasized the intertwined nature of these traits (*Figure 2c*). Despite only taking into account genetic effects, MR largely clustered metabolites in a way that reflects their biochemical groups. The extensive pleiotropy

across the 16 traits, with similar sharing inside biochemical groups, is also illustrated by the structure visible in the normalized effect sizes for each metabolite GWAS hit (*Figure 2d*). Together, these analyses support substantial, but not always consistent, genetic overlap between the traits, particularly in the polygenic components. In the remainder of this article, we will seek a deeper understanding of the biochemical relationships between genotypes and metabolite levels.

## Characterizing the biological functions of candidate genes

An important step in understanding the pathway-level mechanisms of variants is knowing which gene a variant is affecting and how that gene relates to the biology of the pathway. Different types of genes influence trait biology through distinct mechanisms. Metabolite biology is documented in genetic and biochemical databases based on the extensive history of biochemical research (*Supplementary file 2*). Thus, we developed a pipeline for annotating the 213 metabolite GWAS hits with a single most likely gene using gene proximity and manual curation of these databases (*Supplementary file 3* and *Figure 2—figure supplement 4*; see 'Methods'). We annotated 68 variants with genes encoding pathway-relevant enzymes (25-fold enrichment, 95% CI [20-fold, 33-fold], Poisson rate test p<2e-16), 46 with genes encoding transporters (5.2-fold enrichment, 95% CI [3.7-fold, 7.2-fold], p=9e-16), and 30 with genes encoding transcription factors (TFs; 7-fold enrichment among liver marker TFs, 95% CI [3.0-fold, 14-fold], p=3e-5; *Figure 3*). Overall, 69% of variants were assigned to the closest gene and 49% of variants assigned to a pathway-relevant enzyme gene were assigned known IEM genes (*Woidy et al., 2018*). The substantial enrichment for biologically interpretable variants suggests that examining the genetic basis of these traits will allow for the development of hypotheses around relevant molecular mechanisms underlying pleiotropy.

Next we sought to understand which genes and subpathways were most relevant to each biochemical group. We assigned each gene to the biochemical group with the most associated metabolites (*Supplementary file 4*; see 'Methods'). Genes were largely assigned to the group whose relevant biology was nearest the protein encoded by the gene. For example, *BCAT2* encodes an enzyme responsible for the first step in the breakdown of all three BCAAs and was assigned to the BCAA group. *OXCT1* encodes an enzyme responsible for the conversion of acetoacetyl-CoA to the ketone body acetoacetate and was assigned to the Ketone Body group. Similarly, *SLC7A9* encodes a protein that transports amino acids and was assigned to the Other Amino Acid group, while *TCF7L2* is a TF assigned to the Glycolysis group and involved in blood glucose homeostasis. These results confirm that these variants are affecting known trait-relevant biology and reflecting the local structure of these pathways.

Interestingly, a large fraction of the genes involved in trait-relevant biology were genome-wide significant hits for at least one of the 16 metabolites. Specifically, of the 139 total genes encoding enzymes in the pathway diagram for these metabolites (*Figure 3—figure supplement 1*), 51 genes had at least one GWAS hit. Additionally, we performed an ancestry-inclusive GWAS of all 98,189 individuals with complete metabolite data for follow-up analysis. In this ancestry-inclusive analysis, we identified 41 additional hits not found in the European-only GWAS, including associations at 7 additional pathway-relevant genes (*Supplementary file 5* and *Figure 3—figure supplement 2*). This highlights the potential for large-scale, ancestry-inclusive GWAS to discover more biochemically relevant associations among these traits. Together, these findings suggest that GWAS reflect, and have the potential to illuminate, the complex biochemical pathways interconverting these metabolites.

## Investigating the mechanisms of pleiotropy in trait pairs

Given the overlap between the biology of these metabolites and their hits, we next sought to understand the molecular causes of pleiotropy in trait pairs. We found 26 genetically correlated metabolite pairs at a local false sign rate < 0.005. For example, alanine and its strongest genetic correlation partner, isoleucine, share a genetic correlation of $r_g$ = 0.52 (SE = 0.05, p=9e-23). Similarly, plotting the effects of the 213 GWAS variants on these two traits indicates a strong positive correlation (*Figure 4a*). Nonetheless, we noted several outlier loci, including rs370014171 (*PDPR*) and rs77010315 (*SLC36A2*), which have strong discordant effects. We were intrigued to understand why these two variants had discordant effects on alanine and isoleucine relative to their overall positive genetic correlation, while the majority of other variants had concordant effects.

Outlier variants are appealing case studies for understanding the molecular basis of pleiotropy because they affect traits in an exceptional way. Thus, we reasoned that understanding large-effect variants

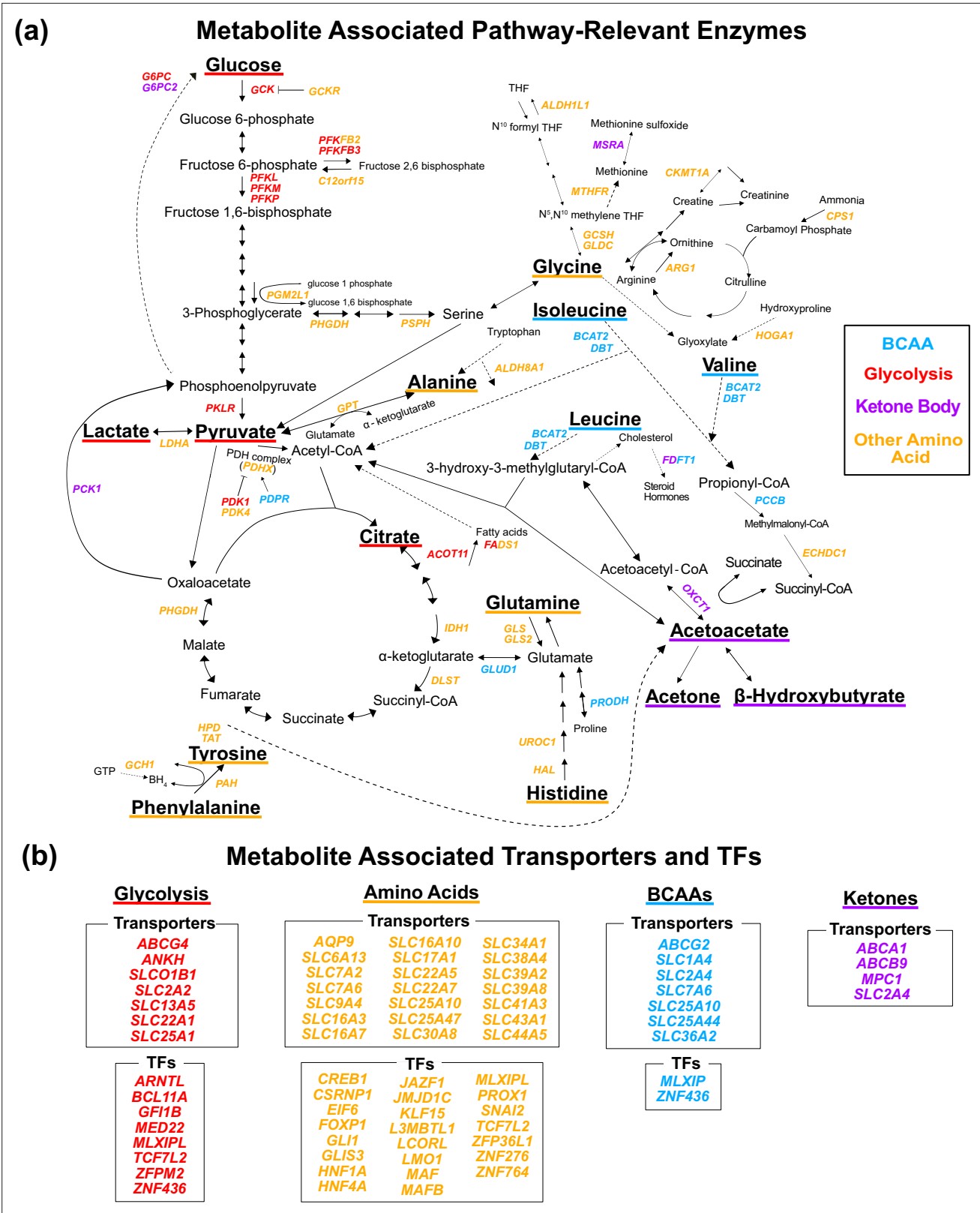

**Figure 3.** Gene annotation of metabolite genome-wide association studies (GWAS) hits. Each gene is colored based on the biochemical group with the most associated metabolites (p<1e-4). If multiple biochemical groups are tied for the most associations for a given gene, they are all shown. (**a**) Expanded pathway diagram with all genes (italicized) that encode pathway-relevant enzymes and were a metabolite GWAS hits. (**b**) List of all genes of the metabolite GWAS hits that encode transporters and transcription factors (TFs). There were 69 metabolite GWAS hits that are not shown. Of these,

*Figure 3 continued on next page*

*Figure 3 continued*

60 were annotated with genes assigned to the gene type general cell function (14 of these 60 were related to lipid function), and 9 were assigned to a gene of unknown function or that did not have any genes nearby (see 'Methods').

The online version of this article includes the following figure supplement(s) for figure 3:

**Figure supplement 1.** Genes in the pathway.

**Figure supplement 2.** Quantile–quantile plots.

inconsistent with the global genetic correlation would reflect interesting biology relevant to the traits. For example, the proteins encoded by *PDPR* and *SLC36A2* are both located between alanine and isoleucine in the biochemical pathway (**Figure 4b**). This suggests that where variants act in the pathway may influence the direction of effect they have on metabolites. To better understand how these two variants affect alanine

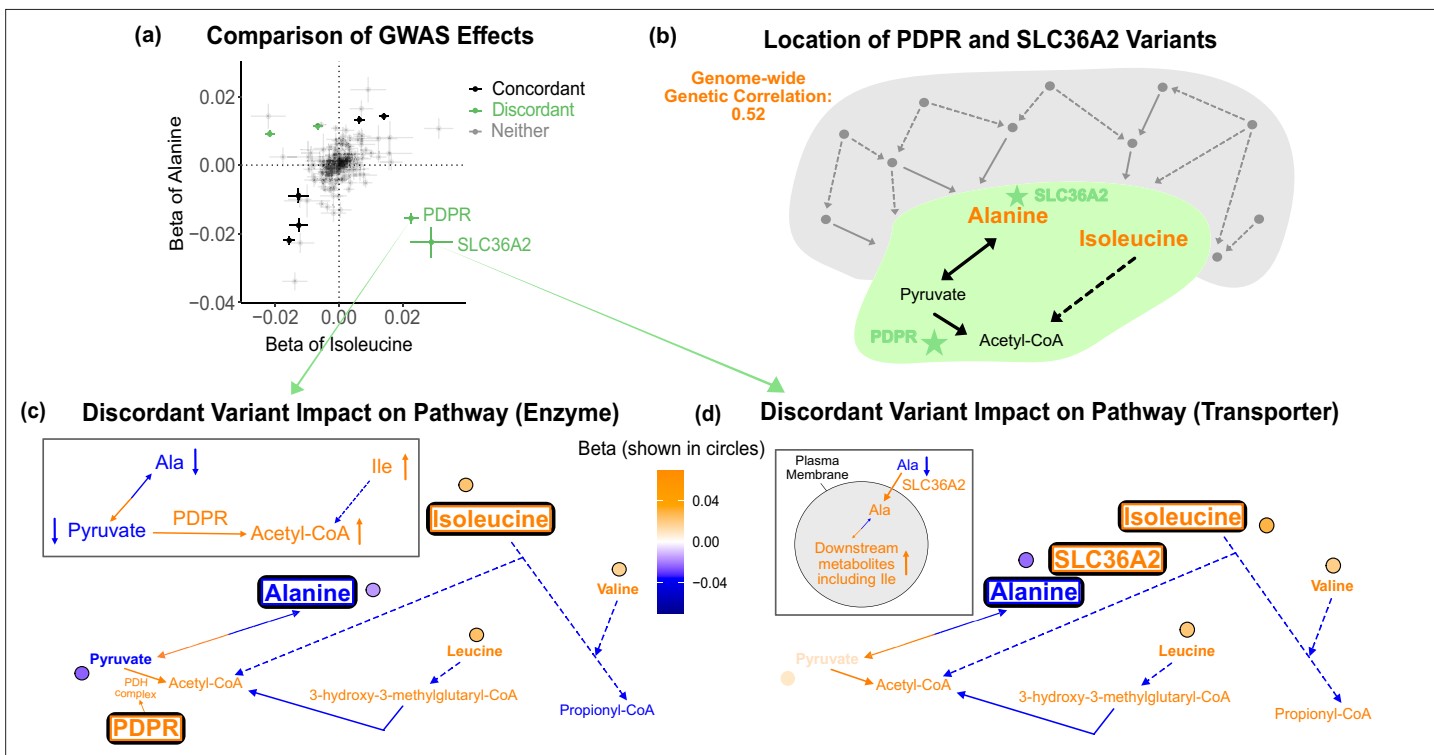

**Figure 4.** Discordant variant analysis. (**a**) Comparison between the effects on alanine levels versus the effects on isoleucine levels for the 213 metabolite genome-wide association studies (GWAS) hits. This highlights two discordant variants: rs370014171 (*PDPR*) and rs77010315 (*SLC36A2*). Variants without an association p<1e-4 in both metabolites are labeled 'Neither.' (**b**) Graphical representation of where these two discordant variants act in the pathway, represented by green stars, relative to other upstream variants driving the positive genetic correlation. Below are the hypothesized mechanisms explaining the GWAS results in relevant metabolites for each of these discordant variants. Data are shown in circles with the coloring corresponding to the effect (beta) of that variant on that metabolite. A black outline represents an association with p<1e-4. Orange text and arrows represent a hypothesized increase (direction, not magnitude) in flux and blue corresponds to a decrease. (**c**) Results for rs370014171 near the gene *PDPR*, which encodes a protein that activates the conversion of pyruvate to acetyl-CoA. All solid lines represent a single chemical reaction step. Dotted lines represent a simplification of multiple steps. (**d**) Results for rs77010315 in the gene *SLC36A2*, which encodes a small amino acid transporter.

The online version of this article includes the following figure supplement(s) for figure 4:

**Figure supplement 1.** Full pathway results for *PDPR*.

**Figure supplement 2.** *PDPR* locus colocalization.

**Figure supplement 3.** Colocalization at *PDPR*.

**Figure supplement 4.** Colocalization at *PDPR* conditioned on secondary signals.

**Figure supplement 5.** Full pathway results for *SLC36A2*.

**Figure supplement 6.** *SLC36A2* locus colocalization.

**Figure supplement 7.** Colocalization at *SLC36A2*.

and isoleucine and explain their outlier behavior, we examined their effect size and direction in the context of their location in the pathway. We then used the variants' metabolite associations to develop candidate mechanisms for how each variant could be jointly influencing the levels of these metabolites.

As an illustration, we first consider variant rs370014171. This variant was assigned to gene *PDPR* because it was the second closest gene, the closest pathway-relevant enzyme, and within 100 kb (12.3 kb to its gene boundaries). PDPR activates the enzyme that catalyzes the conversion of pyruvate to acetyl-CoA (*Figure 4c*, *Figure 4—figure supplement 1*). A candidate mechanism for this variant, supported by the effect size and direction for the 16 metabolites where relevant, is that it increases PDPR activity. There was colocalization of association signals across the five significant metabolites using both conditional SNP-level analyses (*Figure 4—figure supplement 2*) and running coloc once adjusting for secondary signals at alanine (*Figure 4—figure supplement 3*, *Figure 4—figure supplement 4*). This would lead to increased conversion of pyruvate to acetyl-CoA and thus decreased pyruvate (β = -0.023 SDs, SE = 0.003, p=3e-20). To compensate for the subsequent decreased pyruvate levels, there would be increased conversion of alanine to pyruvate causing a decrease in alanine. In response to the increased acetyl-CoA, there would be decreased breakdown of metabolites normally catabolized for its production, including isoleucine, resulting in an increase in isoleucine levels. Thus, this variant has an opposite effect on alanine and isoleucine, despite their overall positive genetic correlation, likely because it affects the activity of an enzyme that acts in the pathway between the pair of metabolites. As expected due to the high correlation between the levels of the three BCAAs, this

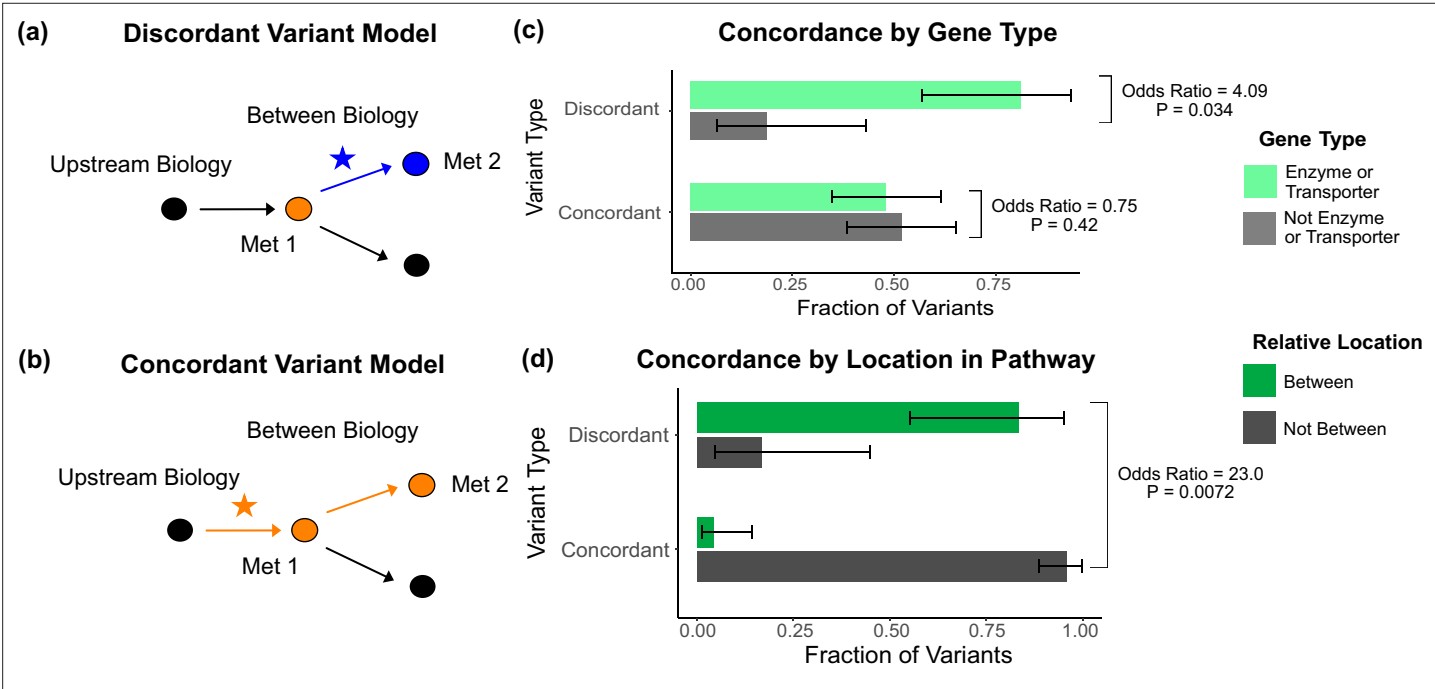

**Figure 5.** Characterization of discordant and concordant variants. (**a**) Proposed model for the mechanism of a discordant variant. This example is for a discordant variant that has opposite effect directions on a pair of metabolites with a positive overall genetic correlation because it affects biology between them. (**b**) Proposed model for the mechanism of a concordant variant. This example is for a concordant variant that has the same effect direction on a pair of metabolites with a positive overall genetic correlation because it affects biology upstream both metabolites. (**c**) Fraction of the discordant and concordant variants that have a pathway-relevant enzyme or transporter gene-type annotation versus those with a different gene-type annotation (N total = 62). Discordant variants are enriched for the gene types of pathway-relevant enzyme or transporter, as would be expected in the model of discordant variants generally affecting biology between metabolites. (**d**) Fraction of the discordant and concordant variants annotated with a pathway-relevant enzyme that affect biology between versus not between their significant metabolite pairs (N total = 17). Significance tests were performed using Fisher's exact method and the plotted SEs are from 95% CI calculated by Wilson score interval.

The online version of this article includes the following figure supplement(s) for figure 5:

**Figure supplement 1.** Additional variant models.

**Figure supplement 2.** Negative genetic correlation variant models.

variant is also a discordant variant for alanine with valine ($r_g$ = 0.51, SE = 0.05, p=2e-21), and alanine with leucine ($r_g$ = 0.49, SE = 0.06, p=1e-16).

As a second example, variant rs77010315 is a missense variant in *SLC36A2*. *SLC36A2* encodes a transporter for small amino acids such as alanine (*Figure 4d*, *Figure 4—figure supplement 5*). There was colocalization of association signals across the four significant metabolites using both conditional SNP-level analyses (*Figure 4—figure supplement 6*) and running coloc (*Figure 4—figure supplement 7*). A candidate mechanism explaining the observed metabolite associations in our data and outlier behavior for this variant is that it increases transport of alanine into cells by SLC36A2. This would result in a decrease in levels of alanine in the blood, but an increase of alanine in cells. This additional intracellular alanine would then allow for increased conversion of alanine to pyruvate, thereby increasing levels of downstream metabolites in the blood, including isoleucine. Thus, this variant has an opposite effect on alanine and isoleucine, despite their overall positive genetic correlation, but in this case because it affects biology between the metabolites at the transporter level.

## Quantifying global properties of molecular pleiotropy

Based on these results, we hypothesized that the two variants described above, and others like them, exhibit outlier behavior because they affect biology *between* the two metabolites (*Figure 5a*). We consider biology 'between' a given pair of metabolites as the shortest biochemical path connecting them, which can include a path converting one metabolite to the other, as well as other scenarios such as those involving colliders. This is because all biochemical reactions, including one-way reactions, can have bidirectional causal relationships due to Le Châtelier's principle (see 'Methods' for details; *Figure 5—figure supplement 1*). Genetic correlation reflects the direction of effect that most associated variants have on two traits. However, when two metabolites are biologically near each other, the region containing 'between' biology is relatively small, such that only a minority of variants directly affect the 'between' region.

Thus, we hypothesized that the genetic correlation of two biologically related metabolites mostly reflects the effects of variants upstream or downstream of the metabolites, masking the effects of those between. We developed an analogous hypothesis that variants affecting biology upstream or downstream of the two metabolites have concordant effects (*Figure 5b*). While less common, the overall genetic correlation for two biologically related metabolites can also be negative due to factors such as feedback loops. In this case, variants acting between the two metabolites would have the same direction of effect on both metabolites, making them discordant with the negative overall genetic correlation (*Figure 5—figure supplement 2*).

To evaluate these models, we defined outliers based on the consistency of their effects with the overall LDSC genetic correlation. If a variant had an effect direction opposite the overall LDSC genetic correlation in at least one significant metabolite pair (p<5e-8 in one, p<1e-4 in the other), it was classified as 'discordant.' For example, a discordant variant for a metabolite pair with a positive genetic correlation would have a negative association in one of the metabolites and a positive association in the other. If a variant had an effect direction consistent with the overall genetic correlation for its significant metabolite pairs, it was classified as 'concordant.' Variants without multiple associations, or where associated traits were not significantly genetically correlated, were classified as 'neither.' In total, of the 62 metabolite GWAS hits that had at least one significant metabolite pair, we found 26 total discordant variant–metabolite pairs across 14 variants (*Supplementary file 6*).

We then investigated overall properties of discordant variants relative to concordant ones. We discovered that discordant variants are more likely to affect genes encoding enzymes and transporters than all other genes types, including TFs, general cell function genes, and those of unknown function (odds ratio = 4.09, 95% CI [1.08, 23.1], p=0.034; *Figure 5c*). This is in contrast to concordant variants, which do not show an enrichment for enzymes and transporters relative to other gene types (odds ratio = 0.75, 95% CI [0.38, 1.48], p=0.42). These observations are consistent with our model that discordant variants tend to affect biology between relevant pairs of metabolites since TFs and general cell function genes generally act outside these metabolic pathways. Thus, they are more likely to affect biology upstream or downstream of both metabolites. In addition, for variants affecting pathway-relevant enzymes, where the location in the pathway that the variant is acting relative to the metabolites is clear, we were able to directly test our hypothesis. We found that discordant variants affecting pathway-relevant enzymes are much more likely to act between, rather than upstream or downstream, the metabolites for which they are discordant (odds ratio = 23.0, 95% CI [1.58, 1510.45], p=0.0072; *Figure 5d*).

We then sought to extend this finding by developing a model contrasting the effects of all variants affecting between versus outside biology at a pathway and genome-wide level (*Figure 6a*). In aggregate, this model predicts that pathways overlapping biology between two metabolites will have a local genetic correlation opposite that of nearby adjacent pathways and that the magnitude of both will exceed that of the global polygenic background. As a case study, we focused on alanine and glutamine, which have a weak positive overall genetic correlation ($r_g$ = 0.16, SE = 0.09, p=0.08; *Figure 6b*; *Figure 6—figure supplement 1*). We then ran BOLT-REML (*Loh et al., 2015*) on variants within 100 kb of genes in each pathway and estimated the corresponding local genetic correlations (see 'Methods').

We found that the local genetic correlations around genes in the Glycolysis, Gluconeogenesis, and Citric Acid Cycle Pathway and around genes in the Other Amino Acid Pathway were negative (*Figure 6c*). Both of these pathways encompass genes affecting biology between alanine and glutamine (*Figure 6d*). In striking contrast, nearby pathways, such as the Urea Cycle, had a positive local genetic correlation for these metabolites ($r_{g,l}$ = 0.45; SE = 0.15, p=0.003). Similarly, we found that regions overlapping genes encoding metabolite-associated transporters and TFs had strong positive genetic correlations consistent with their shared role in the upstream regulation of these two traits ($r_{g,l}$ = 0.44, SE = 0.05, p=1e-20; $r_{g,l}$ = 0.55, SE = 0.06, p=2e-20). All genes outside the core pathways had a weak positive genetic correlation, perhaps reflecting that they are embedded in the global gene regulatory network ($r_{g,l}$ = 0.068, SE = 0.02, p=0.003). Our findings were broadly consistent using individual-level data with Haseman–Elston regression (*Yang et al., 2010*), and summary statistics with $\rho$-HESS (*Shi et al., 2017*), stratified LD score regression (*Bulik-Sullivan et al., 2015a*), and a nonparametric Fligner–Killeen variance test (see 'Methods'; *Supplementary file 7*). These results support the model that variants affecting biology between the metabolites frequently contrast with the contributions of upstream and downstream pathways. This emphasizes that the heterogeneity in genetic effects reflecting local biology shared by the traits can be masked in the global genetic correlation. In addition, these results offer biological intuition for interpreting genetic correlation of molecular traits at a pathway and genome-wide level.

## Using metabolites to understand the mechanism of a disease-associated variant

Motivated by the interpretability of these results, we applied this logic to develop an example model for a variant associated with increased risk for a disease (*Figure 7a*). In this model, we hypothesized one mechanism for how a variant could be associated with increased risk for a disease is that it could impact metabolites in a way that is consistent with disease etiology. For example, the variant could increase metabolites associated with increased risk for the disease and/or decrease metabolites associated with decreased risk.

To apply this model to our data, we considered metabolite GWAS hits that were annotated with pathway-relevant enzymes and associated with increased risk for coronary artery disease (CAD) (*Kichaev et al., 2019*; *Koyama et al., 2020*). The variant that best fit these criteria was rs61791721. This variant was assigned the nearest pathway-relevant enzyme gene, *PCCB*, which encodes a protein that catalyzes the conversion of propionyl-CoA to succinyl-CoA at the intersection of BCAA and fatty acid oxidation (*Figure 7—figure supplement 1*).

We combined results from the literature and incident analysis to understand the association of relevant metabolites with CAD (*Figure 7—figure supplement 2*). We then compared these with the effects of this variant on these metabolites (*Figure 7b*). In this analysis, we included high-density lipoprotein cholesterol (HDL-C), low-density lipoprotein cholesterol (LDL-C), total fatty acids, and total triglycerides due to the extensive evidence implicating their association with CAD and because they are directly adjacent to the biology of the other 16 metabolites. Consistent with the metabolites' corresponding direction of risk for CAD, this *PCCB* variant was negatively associated with glycine and HDL-C, and positively associated with isoleucine, leucine, valine, tyrosine, total fatty acids, total triglycerides, and LDL-C (p<1e-5; *Supplementary files 8 and 9*).

This *PCCB* variant has been associated with CAD in multiple prior GWAS (*Kichaev et al., 2019*; *Koyama et al., 2020*), yet neither the gene this variant affects nor the biological mechanism explaining its association with CAD are known. However, this variant affects many metabolites associated with CAD in a direction consistent with increased risk. Thus, we hypothesized that we could begin to understand why this variant is associated with CAD by understanding the pleiotropic effects of this variant on the metabolites.

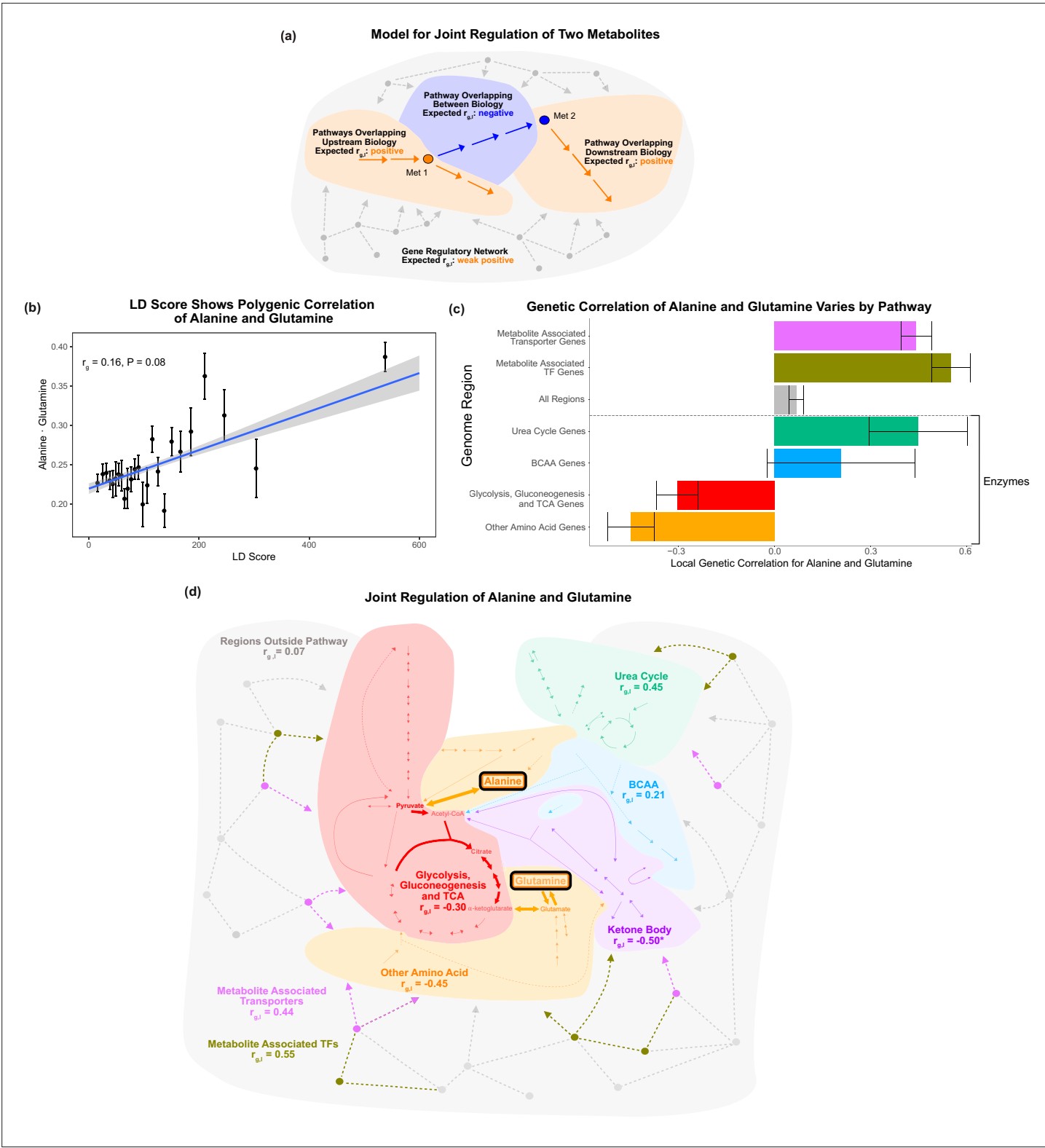

**Figure 6.** Local genetic correlation. (**a**) Model of expected local genetic correlation direction, with contrasting effects of variants affecting 'between' versus outside biology at a pathway and genome-wide level. (**b**) LD Score shows the polygenic correlation of alanine and glutamine. For the x-axis, LD Scores were binned into 25 bins. The y-axis shows the mean and SE within each bin genome wide, and overall genetic correlation significance was calculated using the jackknife standard error (participant N = 94464). (**c**) Results for the local genetic correlation of alanine and glutamine for variants within 100 kb of genes in each pathway. Standard errors are shown (participant N = 94464). Genesets listed below the dotted line include only

*Figure 6 continued on next page*

Figure 6 continued

enzymes and are considered pathway-relevant enzymes for these metabolites. Summary statistics for BOLT-REML and other methods can be found in *Supplementary file 7*. (**d**) Pathway diagram showing the pathways included in the local genetic correlation analysis and the positioning of their genes relative to alanine and glutamine. *Ketone body genes were omitted from panel (**c**) because the limited number of genes meant they failed to robustly converge. All arrows and nodes in the gray section are hypothetical and shown for illustration purposes.

The online version of this article includes the following figure supplement(s) for figure 6:

**Figure supplement 1.** Genetic correlation results.

A potential mechanism that we hypothesized could result in this pathogenic constellation of metabolite effects is that the variant could decrease PCCB activity, resulting in lower levels of succinyl-CoA and increased propionyl-CoA (*Figure 7c*). Consistent with this model, there was colocalization of association signals across all associated traits other than lipids using both conditional SNP-level analyses (*Figure 7—figure supplement 3*) and running `coloc` (*Figure 7—figure supplement 4*). The increased propionyl-CoA could result in excess ammonium being produced, and because alanine is a reservoir for nitrogen waste, this would increase conversion of pyruvate to alanine to capture the toxic ammonium (*Wongkittichote et al., 2017*; *Smith and Garg, 2017*). More glycine may then be broken down in response to the decrease in pyruvate levels, decreasing glycine levels. Conversely, the increased levels of propionyl-CoA would likely mean less valine, isoleucine, fatty acids, and thus triglycerides, would need to be broken down, resulting in an increase in their levels. This increase in fatty acids may stimulate the activity of 3-hydroxy-3-methylglutaryl-CoA (HMG-CoA) reductase and synthase, resulting in an increase in HMG-CoA and cholesterol ;(*Salam et al., 1989*; *Wu et al., 2013*). Increased HMG-CoA could lead to increased leucine because less leucine would need to be broken down to produce HMG-CoA, while increased cholesterol would lead to an increase in LDL-C and a decrease in HDL-C. Therefore, this variant is potentially associated with CAD because it is decreasing PCCB activity, resulting in myriad deleterious downstream metabolic consequences.

While in vivo functional validation would be needed to draw causal conclusions about the effect of this variant on these metabolites and of these metabolites on CAD, this example demonstrates that we can begin to generate informed hypotheses and dissect the molecular basis underlying disease GWAS hits by understanding the mechanism of relevant pleiotropic effects on metabolites. In addition, the pathways implicated by this analysis can also be independently prioritized as potentially playing an important role in cardiometabolic disease by leveraging the molecular basis of genetic correlation discussed in *Figure 6*. For example, alanine and glutamine have opposite associations with CAD and type 2 diabetes despite having an overall positive phenotypic correlation (*Tillin et al., 2015*; *Jauhiainen et al., 2021*). This suggests that the pathways described above with a negative local genetic correlation for alanine and glutamine are likely relevant to the molecular basis of these diseases. Thus, understanding the molecular basis of pleiotropy and genetic correlation of metabolites can improve our understanding of the variants and pathways contributing to complex disease biology.

## Discussion

In this work, we investigate the joint effects of pleiotropic variants on 16 biologically-related metabolites in the context of their biochemical pathways. We build on prior studies examining the genetic architecture of metabolites by characterizing the genes and mechanisms through which variants affect these metabolites, and find a strong enrichment for genes encoding pathway-relevant enzymes and transporters. Our results offer biological intuition for the interpreting genetic correlation of molecular traits at a pathway and genome-wide level.

We demonstrate the effects of variants acting on biology between metabolites often contrast substantially with the contributions of upstream and downstream pathways, as well as the polygenic background. Perhaps paradoxically, while the overall genetic correlation between two traits provides a global view of shared effects, the genes that are directly involved in the traits' core biology are most likely to have divergent effects. We show that one explanation of this is the substantial outlier contributions from variants acting directly between metabolites of interest. We anticipate that further mechanisms, such as context-specific variant effects and differential regulation by peripheral genes, will be discovered in future studies.

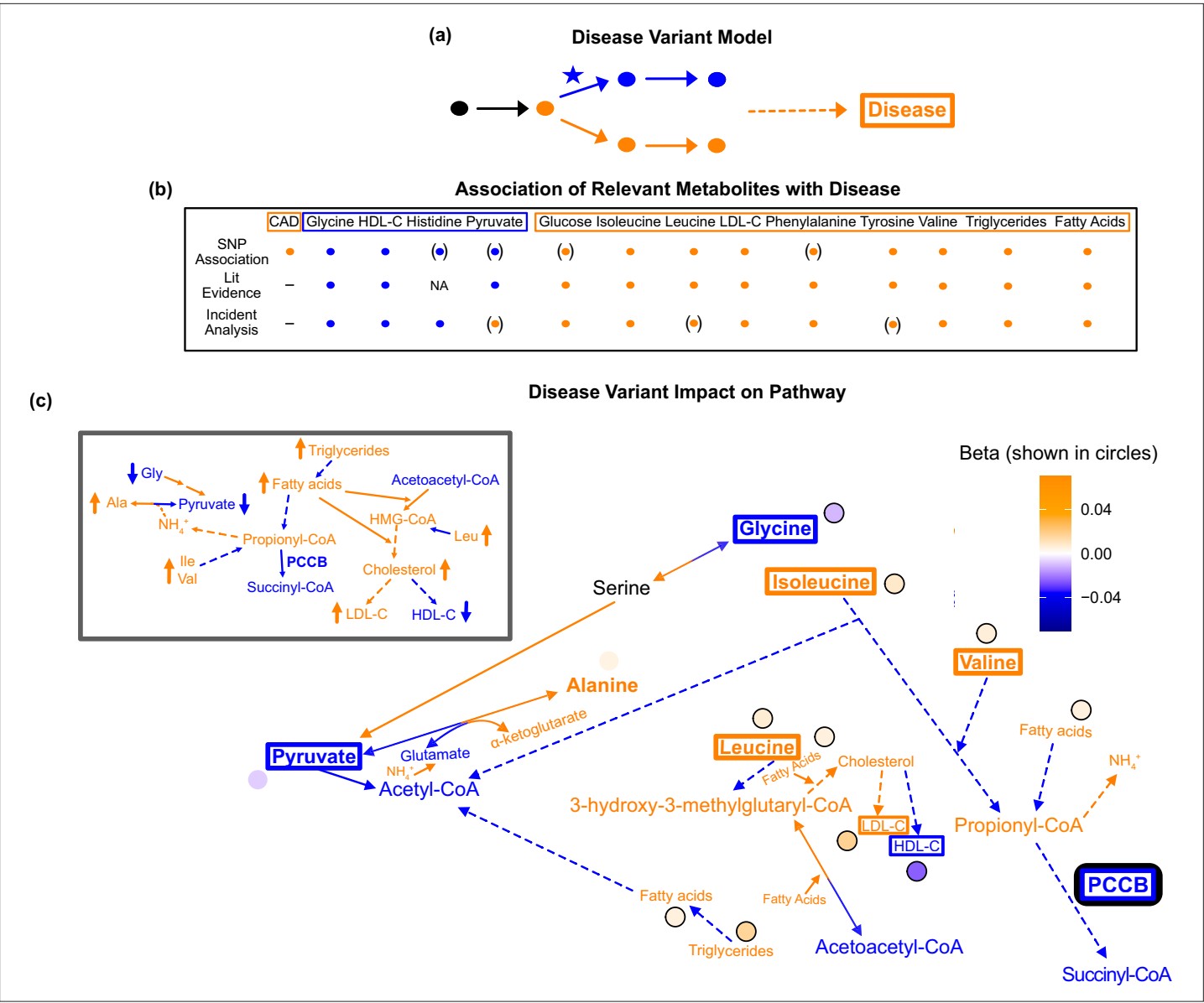

**Figure 7.** Pathway impact and pathology of example disease genome-wide association studies (GWAS) hit. (**a**) Proposed model for the impact of a disease hit on a relevant pathway, contributing to an increased risk in the disease. (**b**) This variant is associated with an increase in levels of metabolites that have been implicated with increased risk of coronary artery disease (CAD), and a decrease in the levels of metabolites that have been implicated with decreased risk. Parentheses indicate nonsignificant associations, 'NA' indicates no evidence was found, and '-' indicates a placeholder because CAD is being compared with itself. (**c**) Results for rs61791721 with gene assignment *PCCB*, which encodes a protein that catalyzes the conversion of propionyl-CoA to succinyl-CoA. The hypothesized mechanism is that the variant is decreasing the activity of PCCB, resulting in the above metabolite associations. Ammonium is represented by its chemical formula ($NH_4^+$). Data are shown in circles with the coloring corresponding to the effect (beta) of that variant on that metabolite. All solid lines represent a single chemical reaction step. Dotted lines represent a simplification of multiple steps.

The online version of this article includes the following figure supplement(s) for figure 7:

**Figure supplement 1.** Full pathway results for *PCCB*.

**Figure supplement 2.** Incident analysis.

**Figure supplement 3.** *PCCB* locus colocalization.

**Figure supplement 4.** Colocalization at *PCCB* for metabolites.

**Figure supplement 5.** rs61791721 eQTL effects.

**Figure supplement 6.** Additional Manhattan plots.

In addition, we show specific examples of candidate molecular mechanisms explaining the association of variants with multiple biologically related metabolites. These include associations at *PDPR*, *SLC36A2*, and *PCCB*, where we show that the direction and magnitude of their effects are consistent with metabolite biochemistry and disease etiology. These proposed molecular mechanisms enable biological prioritization of interesting candidates for future post-GWAS in vitro studies. Overall, these results suggest specific genetic and molecular underpinnings of complex disease variants and provide a road map for further discovery through the interpretation of pleiotropic variant effects on disease-relevant metabolites.

In this work, we focus on metabolites clustered at the intersection of amino acid catabolism, glycolysis, and ketone body metabolism. However, the approaches and results from this article have the potential to reveal novel insights into genetic effects on many biochemical pathways and molecular traits. In addition, integrating this work with proteomic and intermediate metabolomic data will offer additional evidence to develop and support these hypothesized mechanisms. These data may also clarify the relevance of additional mechanisms – such as buffering, feedback, and kinetics – in controlling the plasma levels of these metabolites. Finally, expanding the sample size and diversity of ancestries included in future GWAS, measurements of which are currently underway on the Nightingale platform and others, will increase power to detect novel findings such as important associations for variants with low allele frequency.

While we largely focused on the molecular trait space here, many of these concepts may be useful in the endophenotype and disease space as well. For instance, this approach may help identify variants and pathways most relevant to the core shared biology of a given pair of diseases, potentially revealing more about the molecular bases of the diseases and prioritizing additional drug target candidates. For example, discordant variants have already been identified for some disease pairs, such as for body mass index and type 2 diabetes (*Mahajan et al., 2018*), and non-alcoholic fatty liver disease and type 2 diabetes (*Sliz et al., 2018*).

One limitation of this study is that the metabolites were measured in the blood, while most of the relevant biology and pathways occurs within cells in various tissues throughout the body. Thus, we anticipate extensions of this work to include biomarker measurements from additional cell types and tissues, such as urine, saliva, biopsy samples, and in vitro-differentiated cells. Further, longitudinal analysis of relevant disease cohorts will allow insights into disease progression and subtyping.

In conclusion, this work underscores the potential of unifying biochemistry with genetic data to understand the molecular basis of complex traits and diseases and the mechanism through which variants impact these traits.

## Methods
### Population definition
We defined our GWAS population as a subset of the UK Biobank (*Bycroft et al., 2018*). For our cohort, we use the individuals for which Nightingale plasma metabolite data was available after filtering based on trait QC characteristics (see 'Trait QC and covariate adjustment'). We then filtered individuals by the following QC metrics:

1. Not marked as outliers for heterozygosity and missing rates (`het_missing_outliers` column).
2. Do not show putative sex chromosome aneuploidy (`putative_sex_chromosome_aneuploidy` column).
3. Have at most 10 putative third-degree relatives (`excess_relatives` column).
4. No closer than second-degree relatives.

From these, we defined three cohorts: White British, non-British White, and everyone. We identified White British individuals using the `in_white_British_ancestry_subset` column in the sample QC file. We identified non-British White individuals through self-identification as White, excluding individuals marked as `in_white_British_ancestry_subset` (n = 30,116 who passed QC metrics 1–4 above). As was done for the White British in the initial UK Biobank study design (*Bycroft et al., 2018*), we identified global principal components (PCs) of the genotype data, and then defined ancestry clusters using aberrant with the strictness parameter $\lambda$ = 20. Non-British White individuals who were outliers for any of projected PC pairs PC1/PC2, PC3/PC4, and PC5/PC6 were excluded (n = 25,137 remaining). We performed our first GWAS on the set of individuals in these White British and non-British White cohorts.

The combination of the two sources of European and White British ancestry individuals resulted in a total of 433,390 European ancestry individuals in UK Biobank, of whom 94,464 had available quality-controlled Nightingale data. Our main goal for this study was to understand general principles of genetic architecture, which are not expected to vary among human populations, and thus in the main analysis we excluded non-European individuals on the basis of power and concerns about structure confounding. However, this analysis is significantly limited by the allele frequency differences between populations, and we sought to develop an alternative, inclusive strategy that did not rely on self-identity.

## Metabolomics data generation

The metabolomics data was generated by Nightingale Health using a high-throughput NMR-based platform developed by Nightingale Health Ltd. Randomly selected EDTA non-fasting (average 4 hr since last meal) plasma samples (aliquot 3) from approximately 120,000 UK Biobank participants were measured in molar concentration units. No power calculation was performed, but we anticipated numerous discoveries on the basis of prior GWAS with similar or smaller sample sizes (*Kettunen et al., 2016*; *Willer et al., 2013*; *Lotta et al., 2021*). The measurements took place between June 2019 and April 2020 using six spectrometers at Nightingale Health, based in Finland. The Nightingale NMR biomarker profile contains 249 metabolic measures from each plasma sample in a single experimental assay, including 168 measures in absolute levels and 81 ratio measures. The biomarker coverage is based on feasibility for accurate quantification in a high-throughput manner and therefore mostly reflects molecules with high concentration in circulation, rather than selected based on prior biological knowledge. Additional details about the data generation can be found at here.

## Trait selection and grouping

Sixteen metabolites were chosen from the available Nightingale metabolites based on their biochemical proximity, relevance to health and disease, and because the genes and enzymes involved in their metabolism are well-characterized. Specifically, we first filtered to the 168 metabolites that were not metabolite ratio measurements (n = 81) because we wanted to focus on absolute metabolites levels. We then filtered out the lipids and lipoprotein measures, including cholesterol and fatty acids, because the complexity of their biochemistry make it difficult to map out the chemical reactions directly interconverting one to another, and because many of these metabolites have already been extensively studied in large GWAS (*Willer et al., 2013*; *Graham et al., 2021*). However, many of these are important metabolites in the discussion of cardiometabolic disease so we additionally ran GWAS for total triglycerides, total fatty acids, HDL-C, and LDL-C as part of the interpretation of the *PCCB* variant using the same pipeline as for the 16 metabolites below.

Finally we removed remaining derived measures (such as total combined concentration of BCAA) and those primarily reflecting physiological conditions such as fluid balance (creatinine and albumin) and GlycA (inflammation). One exception to this filtering was the three ketone bodies (3-hydroxybutyrate, acetone, and acetoacetate), which were included due to their proximity and clear direct interconversions connecting them to the metabolic pathways of the remaining amino acid and glycolysis-related metabolites. Metabolites were classified into four biochemical groups based on biochemical similarity. The three branched chain amino acids – valine, leucine, and isoleucine – were classified as "BCAA"; the remaining amino acids in the dataset – glycine, alanine, glutamine, tyrosine, phenylalanine, and histidine – were classified as 'Other Amino Acid'; the three ketone bodies – 3-hydroxybutyrate, acetone, and acetoacetate – were classified as 'Ketone Body'; and the four metabolites in or immediately adjacent to glycolysis – glucose, pyruvate, lactate, and citrate – were classified as 'Glycolysis'.

## Trait QC and covariate adjustment

Trait measurements were filtered to only include baseline samples then log-transformed. Outlier removal was performed by dropping any sample that had a metabolite level greater than 20-fold the interquartile range or greater than 10-fold below the median across all samples for that metabolite. Principal component analysis (PCA) was run for the remaining samples and outliers were dropped using aberrant (lambda = 20) on the top two PCs (*Bellenguez et al., 2012*). Remaining log-transformed measurements were adjusted for spectrometer, week, and weekday. A total of 106,175 individuals had quality-controlled metabolomics data, ranging from 46 to 80 years old (mean 65.5; median 67),

and of whom 54% were female, 90% were genotyped on the UK Biobank array (10% on BiLEVE), 94% identified as White (field 21000 code 1 or 1001–1003), 0.6% identified as multiracial (field 21000 code 2 or 2001–2004), 1.9% identified as Asian or Asian British (field 21000 code 3 or 3001–3004), 1.5% of whom identified as Black or Black British (field 21000 code 4 or 4001–4004), and 1.5% identified with a different label or declined to provide a label (field 21000 code −1, –3, 5, or 6). Samples were subset to the GWAS population defined above, resulting in 94,464 individuals for the European ancestry GWAS.

## Genome-wide association studies

We performed GWAS in BOLT-LMM v2.3.2 (*Loh et al., 2015*) adjusting for sex, array, age, and geno-type PCs 1–10 using the following command (data loading arguments removed for brevity):

```
bolt --phenoCol= [Metabolite] \
        --covarCol=sex \
    --covarCol=Array \
    --qCovarCol=age \
    --qCovarCol=PC{1:10} \
    --lmmForceNonInf \
    --numThreads=24 \
    --bgenMinMAF=1e-3 \
    --bgenMinINFO=0.3
```

The resulting GWAS summary statistics were then filtered to minor allele frequency (MAF) > 0.01 and INFO score >0.7 for further analyses (referred to as the Filtered Metabolite Sumstats). The LDSC munge_sumstats.py script was then used to munge the data (referred to as the Munged Metabolite Sumstats) (*Bulik-Sullivan et al., 2015b*).

## GWAS hit processing

To evaluate GWAS hits, we took the Filtered Metabolite Sumstats and ran the following command using plink version 1.9 (*Chang et al., 2015*):

```
plink --bfile [] --clump [GWAS input file] --clump-p1 1e-4
      --clump-p2 1e-4 --clump-r2 0.01 --clump-kb 1000 --clump-field P_BOLT_
LMM --clump-snp-field SNP
```

We greedily merged GWAS hits across the 16 metabolites located within 0.1 cM of each other and took the SNP with the minimum p-value across all merged lead SNPs. In this way, we avoided potential overlapping variants that were driven by the same, extremely large, gene effects. This resulted in 213 lead GWAS variants, referred to as the metabolite GWAS hits.

## Gene and gene-type annotation

We defined all genes in any Gene Ontology (GO) (*Ashburner et al., 2000*; *Consortium et al., 2021*), Kyoto Encyclopedia of Genes and Genomes (KEGG) (*Kanehisa and Goto, 2000*), or REACTOME MSigDB (*Liberzon et al., 2011*; *Subramanian et al., 2005*) pathway as our full list of putative genes (in order to avoid pseudogenes and genes of unknown function). We initially extended genes by 100 kb (truncating at the chromosome ends) and used the corresponding regions, overlapped with SNP positions, to define SNPs within range of a given gene. Gene positions were defined based on Ensembl 87 gene annotations on the GRCh37 genome build. We then performed manual curation using GeneCards (*Stelzer et al., 2016*) to validate gene assignments and prioritize a single gene per SNP. Gene boundaries for genes encoding pathway-relevant enzymes in KEGG were extended up to 500 kb and assigned to a variant if the gene was biologically relevant to the metabolites the variant was significant in. If there were multiple genes within 100 kb of the variant, then gene assignments were made based on the following priority order: any genes encoding a pathway-relevant enzyme, genes encoding transporters, genes involved in translation/transcription regulation (referred to as TF for brevity), and any genes whose function is known. If there were multiple genes of the same gene type, then the assignment was made based on the relevance of the gene to the metabolites the variant

was significant in, proximity of the gene to the variant, and, if applicable, any additional evidence in the literature (Oxford BIG [*Elliott et al., 2018*] and Open Target Genetics [*Ghoussaini et al., 2021*; *Mountjoy et al., 2021*]). However, even for these cases where there was not high confidence in the exact gene assignment, for instance, because there were multiple genes from the same gene family nearby, the top gene candidates all had the same gene type. Thus, because the major downstream analyses were designed in a way that only the gene type assigned to each variant mattered, the accuracy of the exact gene assignment should not affect the findings. If no genes with known function were within 100 kb of the variant, then the window was extended up to 200 kb. The distance of a variant to a given gene was defined as the number of base pairs from the variant to the closer of the start or end of the gene boundaries or was set to 0 if the variant was within the gene boundaries.

We classified each gene using GeneCards (*Stelzer et al., 2016*) into one of five gene types: pathway-relevant enzyme, transporter, TF, general cell function, and unknown. Genes encoding enzymes that catalyze a reaction in or adjacent to the direct synthesis or degradation of one of the 16 metabolites were defined as pathway-relevant enzymes using manual curation from GO, KEGG, REACTOME, and Stanford's Human Metabolism Map (*Pathways of Human Metabolism Map, 2021*), in addition to GeneCards. Genes encoding known transporters were classified as transporters. Genes involved in translation/transcription regulation were classified as TF. Genes whose function is known but not already classified as a pathway-relevant enzyme, transporter, or TF were classified as 'general cell function.' Genes with unknown function or if there were no genes within 200 kb of the SNP were classified as 'unknown.' See *Supplementary file 3* for each metabolite GWAS hit's gene and gene-type annotations.

Gene-type enrichments were calculated with a Poisson rate test. The baseline was the total of the GWAS hits among the 1.95 Gb of the genome within 100 kb of a gene in any pathway, and the test was performed with the number of GWAS hits within 100 kb of each pathway of interebst. There were 2.8 GWAS hits per megabase within 100 kb of a pathway-relevant enzyme versus 0.1 GWAS hits per megabase among all genes (25-fold enrichment, Poisson rate test p<2e-16). There were 0.58 GWAS hits per megabase within 100 kb of a transporter versus 0.1 GWAS hits per megabase for all genes (5.2-fold enrichment, Poisson rate test p=9e-16). We also repeated this analysis using closest genes rather than assigned genes, which allowed us to use a Fisher's exact test (as each variant has a single closest gene). This resulted in a 27-fold (p<2e-16) enrichment for pathway-relevant enzymes and an 11-fold (p<2e-15) enrichment for transporters, respectively.

For TF enrichments, we used TF-Marker (*Xu et al., 2022*) to annotate tissue-specific marker gene TFs. We considered 'TF' (n = 1316) and 'TFMarker' (n = 18) genes as relevant genes, and TF Pmarker (n = 1424) genes as putatively relevant. We considered enrichment among the 628 genes not associated with cancer or stem cell biology (of which 267 are putative) as our set of tissue-specific TFs for downstream analysis. We consider our background in all cases to be our total GWAS hits number (n = 213) compared to the effective genome size (2.86 Gb). In specifically this TF set, we observed a 5.96-fold enrichment over the genome-wide background (0.44 GWAS hits per megabase, p=4e-6) among relevant gene bodies and 2.50-fold enrichment (0.19 hits/Mb, p=0.0007) within 100 kb of a relevant gene, which was comparable for putatively relevant genes (4.85-fold and 2.86-fold, respectively). This was substantially higher than that of all genes in the genome (1.69-fold within gene bodies and 1.36-fold within 100 kb of genes in any pathway) and comparable to that of all TFs regardless of their function in cancer or stem cells (4.82-fold within gene bodies and 2.18-fold within 100 kb).

We next filtered tissue-specific TFs to those acting in liver (28 relevant and 30 putative), kidney (25 relevant and 20 putative), or pancreas (14 relevant and 3 putative). Kidney and pancreas TFs had no more than one GWAS hit each and were excluded for these analyses. For liver TFs, we observed an 18-fold enrichment (1.3 GWAS hits per megabase, p=0.0057) within gene bodies and a 7.6-fold enrichment (0.56 hits/Mb, p=0.002) within 100 kb of genes. Results were similar when removing the cancer and stem cell filter (1.23 hits/Mb and 0.51 hits/Mb, respectively) and dropped slightly when further including putatively relevant TFs (0.71 hits/Mb and 0.48 hits/Mb). Together, this suggests that liver marker TFs are specifically enriched for variants affecting our metabolite levels.

## Ancestry-inclusive GWAS

For the ancestry-inclusive analysis, we performed the same method as the European-ancestry-only analysis except we omitted the step filtering individuals on the basis of self-identified race/ethnicity and ancestry PC outlier status. For the ancestry-inclusive analysis, we again used the European

ancestry LD matrix as European-ancestry individuals were the overwhelming majority in the study. This resulted in a total of 98,189 individuals for the GWAS, which identified 238 lead GWAS variants across the 16 metabolites. This was inspired by recent 'mega-analysis' studies (*Wojcik et al., 2019*). We examined these associations and identified novel genes by comparing the list of pathway genes in the European-only analysis to those discovered in the ancestry-inclusive analysis.

### Coloc-based colocalization

Full summary statistics for all quality-controlled SNPs within 1 Mb of the target gene were considered at each locus, and these were loaded for all the evaluated traits. Standard deviations of the technical covariate-adjusted trait measurements were used for input standard deviation calculations, and dichotomous traits were set to 'cc'-type datasets while continuous trait were set to 'quant.' `coloc.abf` was run using the default arguments (*Giambartolomei et al., 2014*; *Wallace, 2013*). Output was aggregated and `PP.H4.abf` (the posterior probability of both traits having an association, and that this association is shared) was plotted for all pairs of traits run individually.

### HESS trait heritability and pathway enrichments

We ran HESS (*Shi et al., 2016*) using the following commands:

```
hess.py --local-hsqg {\filtsumstats} --chrom {chrom} \
        --bfile 1kg_eur_1pct_chr{chrom}
        --partition EUR/fourier_ls-chr{chrom}.bed \
        --out {Metabolite}_step1
hess.py --prefix {Metabolite}_step1 --out {Metabolite}_step2
```

where `1kg_eur_1pct_chr{chrom}` were downloaded from here and `EUR/fourier_ls-chr{chrom}.bed` were downloaded from here.

We intersected the resulting heritability estimates per LD block with gene lists from each pathway (see local $\rho$-HESS; within 100 kb of the gene boundary was used as the tested window) and calculated the total heritability within the pathway as the sum of the heritabilities across LD blocks and the variance of the heritability within the pathway as the sum of the variances within each LD block. Overall, this gave a per-pathway estimate. We generated genome-wide estimates of heritability as well as heritability estimates for the subset of the genome nearby any coding gene in MSigDB as background controls from which to estimate the heritability enrichments, and used the coding gene numbers for reporting as they are more conservative.

### LDSC genetic correlation

LD score regression (*Bulik-Sullivan et al., 2015b*) was used to generate genetic correlation estimates. The following command was used:

```
ldsc.py --rg {\mungsumstats} --ref-ld-chr eur_ref_ld_chr
   --w-ld-chr eur_w_ld_chr
```

`eur_*_ld_chr` were downloaded from https://data.broadinstitute.org/alkesgroup/LDSCORE/.

### Mendelian randomization

The Rücker model selection framework was applied. Briefly, MR was run with inverse-variance-weighted (IVW) and MR-Egger with fixed and random effects, and selection between different methods for results to present was based on the goodness-of-fit and heterogeneity parameters for the individual MR regressions as previously described (*Bowden et al., 2018*; *Sinnott-Armstrong et al., 2021c*).

### Discordant variant analysis

All pairwise combinations of LDSC genetic correlation (as described above) were performed for the 16 metabolites. Pairs were filtered to those that had a genetic correlation significantly different than 0 using ashR (*Stephens, 2017*) with a local false sign rate of 0.005. We then annotated all metabolite

GWAS hits with pairs of metabolites for which the variant had a p<1e-4 association with both metabolites and a p<5e-8 association with at least one, defined as significant metabolite pairs. A variant was classified as 'discordant' if it had the same effect direction in both metabolites of at least one significant metabolite pair that had a negative global genetic correlation, or if it had opposite effect directions in the two metabolites of at least one significant metabolite pair that had a positive global genetic correlation. Fourteen variants of the 62 that had at least one significant metabolite pair were classified as discordant. Variants that had the same set of effect directions as the sign of the global LDSC genetic correlation for all of its significant metabolite pairs were classified as 'concordant.' Variants that had no significant metabolite pairs were classified as 'neither'.

The 'between' region for a given pair of metabolites was defined as the shortest realistic biochemical path connecting them, and any alternative paths of reasonably similar distance and likelihood. This region can include a path converting one metabolite to the other, as well as other scenarios such as those involving colliders. This is because all biochemical reactions, including one-way reactions, can have bidirectional causal relationships due to Le Châtelier's principle. In other words, even in a one-way (irreversible) reaction, changes in product levels can induce changes in reactant levels in order to re-establish equilibrium. Genes were defined as acting between a given metabolite pair either if they encoded an enzyme catalyzing a reaction in the 'between' region defined above or if they encoded a transporter that primarily transports either of the two metabolites themselves or an intermediate metabolite in the 'between' region. Variants were defined as acting between a given metabolite pair if the gene they affect was defined as between. Pathways were defined as between a given metabolite pair if many of the genes defined as between the metabolites were part of the pathway or if many of the genes in the pathway were defined as between. Note that even if a pathway is defined as 'between,' not all genes in the pathway will always be between and vice versa; however, this is likely to only make the resulting differences in genetic correlation for 'between' vs. not 'between' pathways more conservative.

## Local $\rho$-HESS

We ran HESS (*Shi et al., 2017*) using the following commands:

```
hess.py --local-rhog {Met1_sumstats}
        {Met2_sumstats} --chrom {chrom} --bfile 1kg_eur_1pct_chr{chrom} \
        --partition EUR/fourier_ls-chr{chrom}.bed --out {Met1_Met2}_step1
\
hess.py --prefix {Met1_Met2}_step1_trait1 \
        --out {Met1_Met2}_step2_trait1
hess.py --prefix {Met1_Met2}_step1_trait2 \
        --out {Met1_Met2}_step2_trait2
hess.py --prefix {Met1_Met2}_step1 \
        --local-hsqg-est {Met1_Met2}_step2_trait1 {Met1_Met2}_step2_trait2
\
        --num-shared 94464 \
        --pheno-cor {gcov_int from LDSC genetic correlation for Met1_Met2}
\
        --out {Met1_Met2}_step3
```

where `1kg_eur_1pct_chr{chrom}` were downloaded here and `EUR/fourier_ls-chr{chrom}.bed` were downloaded here.

We then used the local rho HESS results and estimated the local genetic covariance and correlation across all LD blocks overlapping pathway regions.

We defined the pathway regions based on gene boundaries of relevant genes in *Figure 3—figure supplement 1* as follows: 'Other Amino Acid Genes' includes all genes colored orange, 'Ketone Body Genes' includes all genes colored purple, 'Glycolysis, Gluconeogenesis, and TCA Genes' includes all genes colored red, and 'Urea Cycle Genes' includes all genes colored green. 'BCAA Genes' included all genes in KEGG_VALINE_LEUCINE_AND_ISOLEUCINE_DEGRADATION except *OXCT2*, *HMGCL*, *HMGCS1*, *HMGCS2*, *ACAT1*, *ACAT2*, *OXCT1*, *DLD*, *AGXT2*, *ABAT*, and *AACS* and also included *ECHDC1*. 'All Regions Outside Pathway Genes' was defined as all LD blocks not overlapping any

of the regions defined above. 'Metabolite-Associated TF Genes' and 'Metabolite-Associated Transporters Genes' were defined as all LD blocks overlapping any of TFs or transporters, respectively annotating the metabolite GWAS hits.

### Fligner–Killeen variance test

Rather than aggregating variant effects and estimating total genetic covariance and heritability per pathway, which is not robust to outlier effects, we additionally tried a nonparametric approach. Individual $r_g$ and $h^2$ estimates for LD blocks were compared between the baseline (all coding genes) and the pathway of interest by listing all per-block genetic covariance scores and computing a Fligner–Killeen variance test within each pathway in R. This enables direct evaluation of genetic covariances between the pathways at the cost of simultaneously capturing the enrichment of heritability and genetic covariance therein.

### BOLT-REML

Genotyped variants within 100 kb of genes in each pathway were aggregated, and the resulting matrices were tested using the following command in BOLT-LMM:

```
bolt
    --remove {non-European ancestry individuals}
    --phenoFile={Technical-adjusted metabolites} \
    --phenoCol=Ala \
    --phenoCol=Gln \
    --covarCol=sex --covarCol=Array
    --qCovarCol=age
    --qCovarCol=PC{1:10} \
    --geneticMapFile=genetic_map_hg19_withX.txt.gz '# downloaded with
bolt' \
    --numThreads=24
    --verboseStats \
    --modelSnps {pathway SNPs} \
    --reml \ --noMapCheck
```

Standard errors were as reported by BOLT-REML.

### Haseman–Elston regression

Genotyped variants were pruned to MAF > 1% and approximate linkage equilibrium among individuals included in the GWAS using,

```
{GCTA} --HEreg-bivar {trait1} {trait2} --thread-num 16 --grm {GRM}
```

Results using multiple GRMs (`--mgrm`) to jointly test all pathways were qualitatively similar outside of the genome-wide GRM, which no longer captured the within-pathway component.

### Stratified LD score regression

Analyses were performed as described in LDSC genetic correlation, except that rather than `eur_ref_ld_chr` as the reference LD scores, instead LD scores computed on variants within 100 kb of genes in each pathway were utilized.

### Disease variant analysis

The metabolite GWAS hits annotated with pathway-relevant enzymes were overlapped with significant hits for CAD, identifying the variant rs61791721 as the most significant variant (*Kichaev et al., 2019*; *Koyama et al., 2020*). Incident CAD cases were defined among UK Biobank participants as those individuals who received a first diagnosis of myocardial infarction (MI) using the analytical MI model (field 42000) after the date of baseline assessment. Prevalent cases (individuals with a first diagnosis before date of assessment) were excluded. A Cox proportional hazard model was run with the technical-covariate-adjusted, log-transformed metabolite levels predicting incident MI status,

adjusted for age, age², age * sex, age² * sex, and statin usage (defined based on a list of individual drug codes as previously described; *Sinnott-Armstrong et al., 2021a*). Effect sizes presented are based on the estimates from these models run independently for each metabolite.

## Colocalization analysis

We wanted to evaluate the extent to which our associations might represent single causal variants across multiple traits and used conditional association at the locus to evaluate this. For each variant within 500 kb of our lead SNPs in at least one metabolite, we ran a conditional analysis for the variants within 1 Mb of the gene body of our putative target gene. Then we ran the following association test in plink2:

```
plink2 --glm cols = chrom,pos,ref,alt,a1freq,firth,test,
nobs,orbeta,se,ci,tz,p hide-covar omit-ref
     --pfile<imputed genotypes>
--covar<age/sex/PCs>
--keep<94,464 European-ancestry individuals in the BOLT-LMM GWAS>
--out conditional/$gene/$snp
--pheno<technical-residualized traits>
  --extract <(variants within 1Mb of gene body)
--condition<conditional SNP>
```

For single SNP conditioning tests and `--condition-list` for conditioning on multiple variants. Associations were visually inspected to detect highly linked variants and conditioning tests were repeated with top associations in any of the key traits until there were no significant variants remaining.

For the *PCCB* vignette, additional traits were included in the analysis, including fatty acids and lipids in the Nightingale-assayed individuals and clinical biomarkers in the full cohort of European-ancestry UK Biobank participants, where traits were residualized as previously described (*Sinnott-Armstrong et al., 2021a*). We further included a GWAS for 'hard' CAD as previously defined (*Inouye et al., 2018*), for which results were qualitatively similar when evaluating 'soft' CAD (including angina cases) and employing only EHR-based diagnoses (rather than additionally including self-reported case status). Results for 'hard' CAD are shown in the supplement.

## Pathway diagrams

Diagrams were drawn using Affinity Design, and molecular structures were made using ChemDraw. Pathway information was curated from GO (*Ashburner et al., 2000*; *Consortium et al., 2021*), KEGG (*Kanehisa and Goto, 2000*), or REACTOME MSigDB (*Liberzon et al., 2011*; *Subramanian et al., 2005*), and Stanford's Human Metabolism Map (*Pathways of Human Metabolism Map, 2021*), along with manual curation from public domain biochemistry knowledge (*Supplementary file 2*).

## Acknowledgements

We thank Alyssa Lyn Fortier, Hanna M Ollila, Shoa Clarke, and other members of the Pritchard lab and Assimes lab for helpful discussions. The authors are grateful to UK Biobank and its participants for access to data to undertake this study (Project #30418 and #24983). Nightingale Health Plc is acknowledged for early access to the UK Biobank NMR biomarker data. CJS was supported by a National Science Foundation Graduate Research Fellowship and Stanford's Knight-Hennessy Scholars Program. This work was supported by NIH grants 5R01HG011432 and 5R01AG066490 (to JKP). We would like to thank the reviewers for their careful reading of our manuscript and their thoughtful comments and suggestions that improved our manuscript.

## Additional information

### Competing interests

Anna Cichońska: is a former employee and holds stock options with Nightingale Health Plc. Heli Julkunen: is an employee and holds stock options with Nightingale Health Plc. Eric B Fauman: is

affiliated with Pfizer Worldwide Research, has no financial interests to declare, contributed as an individual and the work was not part of a Pfizer collaboration nor was it funded by Pfizer. Peter Würtz: is an employee and shareholder of Nightingale Health Plc. The other authors declare that no competing interests exist.

### Funding

| Funder | Grant reference number | Author |
|---|---|---|
| Stanford Knight-Hennessy Scholars Program | Graduate Student Fellowship | Courtney J Smith |
| National Science Foundation | Graduate Student Fellowship | Courtney J Smith |
| National Institute of Health | 5R01HG011432 and 5R01AG066490 | Jonathan K Pritchard |

The funders had no role in study design, data collection and interpretation, or the decision to submit the work for publication.

### Author contributions

Courtney J Smith, Conceptualization, Data curation, Formal analysis, Validation, Investigation, Visualization, Methodology, Writing – original draft, Project administration, Writing – review and editing; Nasa Sinnott-Armstrong, Conceptualization, Data curation, Formal analysis, Supervision, Validation, Investigation, Visualization, Methodology, Writing – original draft, Project administration, Writing – review and editing; Anna Cichońska, Heli Julkunen, Data curation, Methodology, Writing – review and editing; Eric B Fauman, Data curation, Investigation, Writing – review and editing; Peter Würtz, Data curation, Supervision, Funding acquisition, Methodology, Writing – review and editing; Jonathan K Pritchard, Conceptualization, Supervision, Funding acquisition, Methodology, Writing – original draft, Project administration, Writing – review and editing

### Author ORCIDs

Courtney J Smith http://orcid.org/0000-0002-7812-0083
Nasa Sinnott-Armstrong http://orcid.org/0000-0003-4490-0601
Heli Julkunen http://orcid.org/0000-0002-4282-0248
Peter Würtz http://orcid.org/0000-0002-5832-0221
Jonathan K Pritchard http://orcid.org/0000-0002-8828-5236

### Ethics

Human subjects: All participants provided written informed consent and ethical approval was obtained from the North West Multi-Center Research Ethics Committee (11/NW/0382). The current analysis was approved under UK Biobank Project 24983 and 30418.

### Decision letter and Author response

Decision letter https://doi.org/10.7554/eLife.79348.sa1
Author response https://doi.org/10.7554/eLife.79348.sa2

# Additional files

### Supplementary files

• Supplementary file 1. Metabolite HESS heritabilities. Heritability results from HESS for each metabolite.

• Supplementary file 2. Gene function sources. Sources for biochemical characterization of genes mentioned in the variant vignettes.

• Supplementary file 3. Metabolite genome-wide association studies (GWAS) hits annotation. Annotation for the 213 metabolite GWAS hits, including the assigned gene, assigned gene type, variant classification, and nearest gene.

• Supplementary file 4. Gene biochemical groups. The number of significant (p<1e-4) metabolite associations each metabolite genome-wide association studies (GWAS) gene had for each biochemical group. For a given gene, only biochemical groups that had at least one significant

metabolite association were listed.

• Supplementary file 5. Ancestry-inclusive genome-wide association studies (GWAS) hits. List of additional metabolite GWAS hits from the ancestry-inclusive analysis that were not present in the European-only GWAS results.

• Supplementary file 6. Discordant variant annotation. List of each discordant variant–metabolite association, including the variant annotations and relevant metabolite pair genetic correlation and genome-wide association studies (GWAS) summary statistics.

• Supplementary file 7. Local genetic correlation results. Combined results for different methods of calculating the local genetic correlation for different pathways for alanine and glutamine, demonstrating the consistency across the different approaches.

• Supplementary file 8. Metabolite associations with coronary artery disease (CAD). Literature evidence and citations for metabolite associations with CAD.

• Supplementary file 9. *PCCB* genome-wide association studies (GWAS) results. GWAS summary statistics for rs61791721 (*PCCB*) in the 16 metabolites.

• MDAR checklist

### Data availability

The source data and analyzed data have been deposited in Dryad at https://doi.org/10.5061/dryad.79cnp5hxs. Code are available at the github link (https://github.com/courtrun/Pleiotropy-of-UKB-Metabolites copy archived at swh:1:rev:bff4f4d8fb0562f222b1f73560b23ca9b8f57047). The raw individual level data are available through application to UK Biobank.

The following dataset was generated:

| Author(s) | Year | Dataset title | Dataset URL | Database and Identifier |
|---|---|---|---|---|
| Smith C, Sinnott-Armstrong N, Cichonska A, Julkunen H, Fauman E, Wurtz P, Pritchard J | 2022 | Pleiotropy of UK Biobank Metabolites [preliminary] | https://doi.org/10.5061/dryad.79cnp5hxs | Dryad Digital Repository, 10.5061/dryad.79cnp5hxs |

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
