## [Editor Report]

Smith and colleagues provide a framework for understanding a seemingly paradoxical observation in human genetics: two phenotypes may be closely correlated to each other, and the patterns of genetic variation that influence both phenotypes may be widely shared at the genome-wide level, but there are often specific genetic variants that show discordant patterns. Though the observations in this article are derived from analysis of metabolic phenotypes, this may have broader relevance to interpreting the results from disease-related genetic association studies, and shed light on the processes that connect different disease phenotypes.

---

## [Decision Letter]

**Decision letter after peer review:**

Thank you for submitting your article "Integrative analysis of metabolite GWAS illuminates the molecular basis of pleiotropy and genetic correlation" for consideration by *eLife*. Your article has been reviewed by 3 peer reviewers, including Alexander Young as the Reviewing Editor and Reviewer #1, and the evaluation has been overseen by George Perry as the Senior Editor. The following individual involved in the review of your submission has agreed to reveal their identity: Mark I McCarthy (Reviewer #2).

Essential revisions:

The reviewers all agreed that your article presented interesting and potentially important results about how to link GWAS results to biochemical mechanisms. However, the reviewers have raised issues with the presentation that need to be addressed before the manuscript can be accepted. Furthermore, please address reviewer #3's comment about colocalization, and reviewer #1's comment about the ancestry-inclusive GWAS.

*Reviewer #1 (Recommendations for the authors):*

I am quite confused as to where the ancestry inclusive analysis was used? It is mentioned in the introduction as identifying additional hits, but are these used anywhere in the main text results? I also found the details on the ancestry inclusive analysis in the methods lacking. Was the same GWAS procedure used as for the European ancestry GWAS? Did the results look reasonable in terms of QQ-plots, etc.?

For the model proposed in Figure 5a, this seems quite different from the example given in Figure 4c for the variant affecting PDPR. For the variant affecting PDPR, it appears that this discordant effect is generated due to the variant affecting the conversion of pyruvate into Acetyl-CoA, and the corresponding increase in conversion of alanine into pyruvate, and a decrease in the conversion of BCAAs into Acetyl-CoA. So the key here seems to be the level of something (Acetyl-CoA) that is downstream of both of the pair of metabolites. The model in Figure 5a, in contrast, appears to be implying that discordant effects are generated by variants affecting the pathway leading to the conversion of one metabolite into another. So is the PDPR example different from the model proposed in Figure 5a? It does not appear to be consistent with the definition of 'between' biology as being a short pathway converting one metabolite to another, but perhaps I am misunderstanding.

I also found the description of the mechanism in Figure 4d a little confusing. isoleucine is described as 'downstream' of alanine (line 185), but the diagram in Figure 4d does not appear to show that.

For the discussion of the putative mechanism linking the PCCB variant with CAD: I think you need to be more careful about whether you are talking about correlation or causation when you talk about a 'pathogenic constellation of metabolite effects'. Many of the correlations between metabolites and CAD risk may not reflect the causal effects of those metabolites on CAD.

Figure 4a: what is the regression line? Did it adjust for sampling error for effects on both phenotypes?

Figure 4b: the position of the 'star' SLC36A2 I found a little ambiguous.

Figure 5 panels c and d: some of the confidence intervals go into negative territory. I don't think it's critical for the paper, but you could try alternative ways of constructing the confidence interval for a binomial proportion, such as a Wilson Score Interval, that should not have this issue.

Often point estimates are given along with P-values (e.g. for genetic correlations), but no confidence interval is given. Please also give confidence intervals.

Line 382: "The included participants are therefore meant to be representative of the 502,543 participants in the full cohort." How is this known?

Line 411: I think alanine is missing from the list

[Editors' note: further revisions were suggested prior to acceptance, as described below.]

Thank you for resubmitting your work entitled "Integrative analysis of metabolite GWAS illuminates the molecular basis of pleiotropy and genetic correlation" for further consideration by *eLife*. Your revised article has been evaluated by George Perry (Senior Editor) and a Reviewing Editor.

Please address the remaining comment from reviewer 1.

*Reviewer #1 (Recommendations for the authors):*

With respect to my comment about the definition of 'between' biology, I thank the authors for the inclusion of Figure 5 – Supplement 1, which is helpful in tying this to the example to Figure 4c. However, I'm still not convinced that the example given in Figure 5 – Supplement 1a is consistent with the definition in the main text: 'We consider biology "between" a given pair of metabolites as the shortest realistic biochemical path converting one to the other'. The example in Figure 5 – Supplement 1a appears to show two metabolites both being converted into something else, rather than one being converted into the other. In the language of directed acyclic graphs, Figure 5 Supplement 1a shows a 'collider' and Figure 5 Supplement 1b shows a 'chain', but the main text definition appears to only consider the 'chain' model. Some further clarification on this in the main text is warranted.

---

## [Author Response]

Essential revisions:The reviewers all agreed that your article presented interesting and potentially important results about how to link GWAS results to biochemical mechanisms. However, the reviewers have raised issues with the presentation that need to be addressed before the manuscript can be accepted. Furthermore, please address reviewer #3's comment about colocalization, and reviewer #1's comment about the ancestry-inclusive GWAS.

Thank you for the feedback. We have now made these changes as well as the additional ones below. See responses to individual reviewer comments below and the document with tracked changes for more specifics.

Reviewer #1 (Recommendations for the authors):I am quite confused as to where the ancestry inclusive analysis was used? It is mentioned in the introduction as identifying additional hits, but are these used anywhere in the main text results? I also found the details on the ancestry inclusive analysis in the methods lacking. Was the same GWAS procedure used as for the European ancestry GWAS? Did the results look reasonable in terms of QQ-plots, etc.?

Thank you for mentioning this. We have now made the following changes to improve the clarity around this analysis and answer your questions:

1. We have removed mention of the ancestry-inclusive analysis from the introduction and instead only discussed it during a dedicated paragraph in the main text results (lines 139-144). We made this change to make it clear that the majority of the analysis and results are from the European-ancestry only GWAS as these are less susceptible to confounding by population structure. However, we still ran and included the ancestry-inclusive GWAS to evaluate the utility of GWAS on diverse ancestries in discovering additional biologically important hits. In the main text we directly mention the number of additional hits identified by the ancestry-inclusive analysis and in a supplementary table (Supplementary File 5) we list the SNPs and their respective summary statistics from each of the GWAS for the associations at additional pathway-relevant genes.

2. We have now created a dedicated section in the methods called “Ancestry-Inclusive GWAS” to describe the details of the method (lines 544-552) and clarify that the method was identical to that of the European-ancestry only GWAS, other than the omission of a single filtering step (the step filtering individuals on the basis of self-identified race/ethnicity and ancestry PC outlier status).

3. We have now added an additional supplementary figure (Figure 3 —figure supplement 2) with QQ-plots for each of the 16 metabolite ancestry-inclusive GWAS. These results and the overlap in these summary statistics and hits with those from the European-ancestry only GWAS support that the ancestry-inclusive GWAS results are reasonable. Please note that for the ancestry-inclusive GWAS we opted to only focus on variant discovery and use the largest European ancestry for methods examining sub-threshold variants to minimize the risk of population structure confounding in those analyses (Meier et al. and Berg et al. height papers, *eLife* 2019).

For the model proposed in Figure 5a, this seems quite different from the example given in Figure 4c for the variant affecting PDPR. For the variant affecting PDPR, it appears that this discordant effect is generated due to the variant affecting the conversion of pyruvate into Acetyl-CoA, and the corresponding increase in conversion of alanine into pyruvate, and a decrease in the conversion of BCAAs into Acetyl-CoA. So the key here seems to be the level of something (Acetyl-CoA) that is downstream of both of the pair of metabolites. The model in Figure 5a, in contrast, appears to be implying that discordant effects are generated by variants affecting the pathway leading to the conversion of one metabolite into another. So is the PDPR example different from the model proposed in Figure 5a? It does not appear to be consistent with the definition of 'between' biology as being a short pathway converting one metabolite to another, but perhaps I am misunderstanding.I also found the description of the mechanism in Figure 4d a little confusing. isoleucine is described as 'downstream' of alanine (line 185), but the diagram in Figure 4d does not appear to show that.

Thank you for your feedback on this. We agree that the “between” concept for more complicated examples like that in Figure 4c and 4d may be confusing to readers, especially those used to looking at gene regulatory networks where the concept of downstream/upstream means something different than that in pathways of chemical reactions. To address this, we have now added an additional supplementary figure (Figure 5 —figure supplement 1) with two additional example models. The first model is based on the metabolite pathway in Figure 4c and 4d and more explicitly defines the regions of between biology vs outside biology (Figure 5 —figure supplement 1a), as well as gives an example of a discordant (Figure 5 —figure supplement 1b) and concordant model (Figure 5 —figure supplement 1c). The second model is an example of a variant affecting a gene encoding a transporter and more explicitly defines the regions of between biology vs outside biology (Figure 5 —figure supplement 1d), as well as gives an example of a discordant (Figure 5 —figure supplement 1e) and concordant model (Figure 5 —figure supplement 1f) for that situation.

For the discussion of the putative mechanism linking the PCCB variant with CAD: I think you need to be more careful about whether you are talking about correlation or causation when you talk about a 'pathogenic constellation of metabolite effects'. Many of the correlations between metabolites and CAD risk may not reflect the causal effects of those metabolites on CAD.

This is an important point and definitely something we agree with. We have now modified the language in the “Using metabolites to understand the mechanism of a disease-associated variant” Results section to more clearly describe the analysis as a proof-of-concept analysis in terms of hypothesis generation and a new way of thinking about these kinds of data, as opposed to implying causality between the PCCB variant or relevant metabolites and CAD. We have also softened the language in the section, replacing many “would”s with “could”s. For example, the phrase you mentioned now reads “A potential mechanism that we hypothesized could result in this pathogenic constellation of metabolite effects is that the variant could decrease PCCB activity…”. We also added a sentence explicitly warning against causal conclusions being drawn: “… in vivo functional validation would be needed to draw causal conclusions about the effect of this variant on these metabolites and of these metabolites on CAD…”.

Figure 4a: what is the regression line? Did it adjust for sampling error for effects on both phenotypes?

Thank you for pointing this out. We have decided to remove the regression line from Figure 4a altogether because it may confuse readers and isn’t explicitly used in our results.

Figure 4b: the position of the 'star' SLC36A2 I found a little ambiguous.

Thank you for mentioning this, we have now modified the location of the star in Figure 4b to more clearly represent that it is a transporter of alanine.

Figure 5 panels c and d: some of the confidence intervals go into negative territory. I don't think it's critical for the paper, but you could try alternative ways of constructing the confidence interval for a binomial proportion, such as a Wilson Score Interval, that should not have this issue.

Thank you for the suggestion to use an alternative way of constructing the confidence interval for the data shown in Figure 5c and 5d. We have now updated the figure to show the 95% confidence interval calculated by Wilson Score Interval, which you were correct did not have the issue of going into negative values.

Often point estimates are given along with P-values (e.g. for genetic correlations), but no confidence interval is given. Please also give confidence intervals.

Thank you, we have now added SE for all the genetic correlation values and GWAS summary statistics, and 95% confidence intervals for the Fisher's exact tests and enrichment analyses mentioned in the text.

Line 382: "The included participants are therefore meant to be representative of the 502,543 participants in the full cohort." How is this known?

This is a good point that just because the participants were randomly selected to undergo metabolomic analysis, that does not mean the random sample will be representative for all relevant characteristics. We have now removed this sentence from the text. In case you are interested, a table comparing some basic characteristics of the ~120k randomly selected participants vs the full ~500k cohort is available in Nightingale’s recent medRxiv paper describing the measurements (Table 1 in https://www.medrxiv.org/content/10.1101/2022.06.13.22276332v2).

Line 411: I think alanine is missing from the list

Thank you for catching this. We have now added alanine to the list.

[Editors' note: further revisions were suggested prior to acceptance, as described below.]

Please address the remaining comment from reviewer 1.Reviewer #1 (Recommendations for the authors):With respect to my comment about the definition of 'between' biology, I thank the authors for the inclusion of Figure 5 – Supplement 1, which is helpful in tying this to the example to Figure 4c. However, I'm still not convinced that the example given in Figure 5 – Supplement 1a is consistent with the definition in the main text: 'We consider biology "between" a given pair of metabolites as the shortest realistic biochemical path converting one to the other'. The example in Figure 5 – Supplement 1a appears to show two metabolites both being converted into something else, rather than one being converted into the other. In the language of directed acyclic graphs, Figure 5 Supplement 1a shows a 'collider' and Figure 5 Supplement 1b shows a 'chain', but the main text definition appears to only consider the 'chain' model. Some further clarification on this in the main text is warranted.

Thank you for this important feedback! We have now updated the definition in the main text (lines 201-205) to read “We consider biology "between" a given pair of metabolites as the shortest biochemical path connecting them, which can include a path converting one metabolite to the other, as well as other scenarios such as those involving colliders. This is because all biochemical reactions, including one-way reactions, can have bi-directional causal relationships due to Le Châtelier's Principle (see Methods for details; Supplementary Figure 5 —figure supplement 1).”

We also updated the expanded definition given in the methods section (lines 605-611) to “The ``between'' region for a given pair of metabolites was defined as the shortest realistic biochemical path connecting them, and any alternative paths of reasonably similar distance and likelihood. This region can include a path converting one metabolite to the other, as well as other scenarios such as those involving colliders. This is because all biochemical reactions, including one-way reactions, can have bi-directional causal relationships due to Le Châtelier's Principle. In other words, even in a one-way (irreversible) reaction, changes in product levels can induce changes in reactant levels in order to re-establish equilibrium.”